# A secretory pathway kinase regulates sarcoplasmic reticulum Ca$^{2+}$ homeostasis and protects against heart failure

Adam J Pollak[1], Canzhao Liu[2], Aparna Gudlur[3], Joshua E Mayfield[1], Nancy D Dalton[2], Yusu Gu[2], Ju Chen[2], Joan Heller Brown[1], Patrick G Hogan[3,4,5], Sandra E Wiley[1], Kirk L Peterson[2], Jack E Dixon[1,6,7]*

[1]Department of Pharmacology, University of California, San Diego, San Diego, United States; [2]Department of Medicine, University of California, San Diego, San Diego, United States; [3]Division of Signaling and Gene Expression, La Jolla Institute for Allergy and Immunology, San Diego, United States; [4]Program in Immunology, University of California, San Diego, San Diego, United States; [5]Moores Cancer Center, University of California, San Diego, San Diego, United States; [6]Department of Cellular and Molecular Medicine, University of California, San Diego, San Diego, United States; [7]Department of Chemistry and Biochemistry, University of California, San Diego, San Diego, United States

**Abstract** Ca$^{2+}$ signaling is important for many cellular and physiological processes, including cardiac function. Although sarcoplasmic reticulum (SR) proteins involved in Ca$^{2+}$ signaling have been shown to be phosphorylated, the biochemical and physiological roles of protein phosphorylation within the lumen of the SR remain essentially uncharacterized. Our laboratory recently identified an atypical protein kinase, Fam20C, which is uniquely localized to the secretory pathway lumen. Here, we show that Fam20C phosphorylates several SR proteins involved in Ca$^{2+}$ signaling, including calsequestrin2 and Stim1, whose biochemical activities are dramatically regulated by Fam20C mediated phosphorylation. Notably, phosphorylation of Stim1 by Fam20C enhances Stim1 activation and store-operated Ca$^{2+}$ entry. Physiologically, mice with *Fam20c* ablated in cardiomyocytes develop heart failure following either aging or induced pressure overload. We extended these observations to show that non-muscle cells lacking Fam20C display altered ER Ca$^{2+}$ signaling. Overall, we show that Fam20C plays an overarching role in ER/SR Ca$^{2+}$ homeostasis and cardiac pathophysiology.
DOI: https://doi.org/10.7554/eLife.41378.001

*For correspondence:
jedixon@ucsd.edu

Competing interests: The authors declare that no competing interests exist.

## Introduction

Heart failure remains the leading cause of death in the United States (*Benjamin et al., 2018*) and is clinically characterized by reduced cardiac function and pathological remodeling. Intracellular calcium (Ca$^{2+}$) handling is the basis of cardiac contraction and relaxation (*Bers, 2014*). Contraction involves the release of sarcoplasmic reticulum (SR) Ca$^{2+}$ into the cytosol via the ryanodine receptor 2 (RyR2); relaxation occurs with SR Ca$^{2+}$ uptake by the sarco/endoplasmic reticulum Ca$^{2+}$ ATPase pump (SERCA2a) (*Bers, 2002*). Multiple proteins and post-translational modifications tightly regulate this SR Ca$^{2+}$ cycling, including phosphorylation of proteins on the cytosolic face of the SR (*Ai et al., 2005*; *Anderson et al., 2011*; *Chaanine and Hajjar, 2011*), and are a focus of several clinical attempts to improve cardiac function (*Marks, 2013*; *Hulot et al., 2016*). In addition, endoplasmic reticulum (ER) Ca$^{2+}$ signaling underlies a broad range of cellular process in a variety of tissues (*Raffaello et al., 2016*). Importantly, the biochemical and physiological roles of protein

phosphorylation within the ER/SR remains essentially unknown. Here we describe a luminal ER/SR kinase that is important for cellular $Ca^{2+}$ signaling and cardiac pathophysiology.

While studies of protein phosphorylation focus, almost exclusively, on the role of kinases in the nucleus and cytoplasm of cells, many important secretory proteins are also phosphorylated (*Zhou et al., 2009*). For over 100 years the identification of the physiological kinase that phosphorylates secretory proteins, such as β-casein, the most abundant phosphoprotein in milk, remained a mystery (*Tagliabracci et al., 2013*). The discovery of the atypical kinase Fam20C solved this conundrum, as Fam20C has a signal peptide that brings it to the lumen of the secretory pathway where it preferentially phosphorylates Ser within Ser-x-Glu/pSer (S-x-E) motifs (*Tagliabracci et al., 2012*). Fam20C is important for bone and teeth formation in patients (*Simpson et al., 2007*; *Cui et al., 2015*) and animal models (*Vogel et al., 2012*). Subsequent studies demonstrated that Fam20C is responsible for generating >90% of the secreted phosphoproteome and that its substrates are likely involved in diverse biological functions that mostly remain uncharacterized (*Tagliabracci et al., 2015*).

Fam20C's potential role in cardiac function was highlighted by the identification of a heart disease-associated polymorphism in luminal SR resident histidine-rich $Ca^{2+}$-binding protein (HRC) (*Arvanitis et al., 2008*; *Singh et al., 2013*). Specifically, this polymorphism (HRC-Ser96Ala) was observed within a Fam20C consensus S-x-E phosphorylation motif and Fam20C was responsible for HRC-Ser96 phosphorylation (*Pollak et al., 2017*). This phosphorylation modulated HRC's interactions with triadin and SERCA2a and was demonstrated to be necessary to prevent arrhythmias in vitro. However, Fam20C's physiological role in normal and diseased heart function remains unknown. Importantly, we hypothesize that several SR $Ca^{2+}$ regulatory proteins are Fam20C substrates, given the extent of SR proteins containing S-x-E motifs and that Fam20C has been shown to be responsible for the majority of secretory pathway phosphorylation in several cell types (*Tagliabracci et al., 2015*).

This study identifies, mechanistically, several critical ER/SR proteins as substrates of Fam20C, and demonstrates Fam20C's broad impact on cellular $Ca^{2+}$ signaling in cardiomyocytes and other cell types. Notably, we show that stromal interaction molecule 1 (Stim1), which is important for controlling cellular $Ca^{2+}$ content in many cell types (*Collins et al., 2013*), and calsequestrin 2 (CSQ2), the major $Ca^{2+}$ binding protein in the SR, are dramatically regulated by Fam20C phosphorylation. We then developed a cardiomyocyte-specific *Fam20c* conditional knockout (cKO) mouse model to determine the physiological roles of SR luminal protein phosphorylation. Upon numerous pathological stimuli, we demonstrate that Fam20C deficiency in cardiomyocytes results in dilated cardiomyopathy (DCM). DCM involves left ventricle (LV) dilation and reduced functional capacity in both human and animal models (*Hershberger et al., 2013*). A better fundamental understanding of the underlying molecular biologic mechanisms of these processes is critical to improving patient outcomes.

## Results

### Fam20C phosphorylates multiple SR regulatory proteins important for $Ca^{2+}$ homeostasis

Given the dramatic biochemical effects of Fam20C phosphorylation of HRC-Ser96 (*Pollak et al., 2017*), we postulated that Fam20C phosphorylation of other SR proteins would have important regulatory functions. Therefore, we sought to uncover new Fam20C substrates to determine the overall mechanistic role of Fam20C phosphorylation in the SR lumen. Focusing on luminal SR proteins that have been shown to regulate $Ca^{2+}$ signaling in cardiac function, we identified several proteins that contain S-x-E motifs and are, therefore, potential Fam20C substrates; this includes calsequestrin 2 (CSQ2) (*Houle et al., 2004*), stromal interaction molecule 1 (Stim1) (*Collins et al., 2013*), calumenin (*Sahoo and Kim, 2010*), sarcalumenin (*Yoshida et al., 2005*), calreticulin (*Wang et al., 2012*), and triadin (*Terentyev et al., 2005*).

We engineered triadin, calumenin, calreticulin, and sarcalumenin, each with a C-terminal FLAG-epitope tag, and co-expressed each of them with HA-epitope tagged Fam20C WT or a catalytically inactive Fam20C mutant, D478A, referred to as Fam20C kinase-inactive (KI) (*Tagliabracci et al., 2012*), in HEK293 cells, followed by $^{32}P$-orthophosphate labeling (*Figure 1A*). Following Flag-immunoprecipitation of cell extracts, immunoblotting, and autoradiography, each of these proteins was

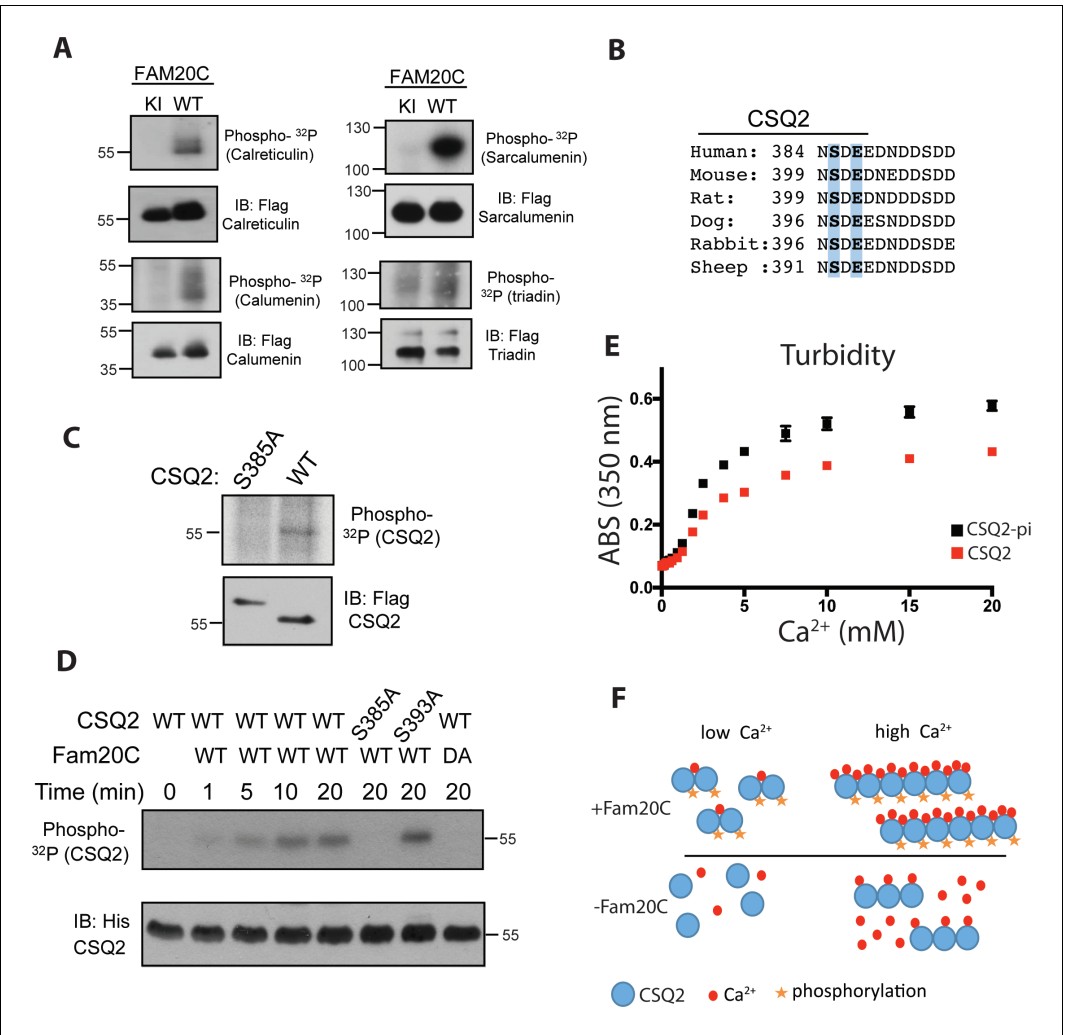

**Figure 1.** Fam20C phosphorylates luminal SR proteins and regulates CSQ2 polymerization. (A) Representative autoradiograph and Flag immunoblot (IB) of Flag immunoprecipitates (IPs) of [32P]-orthophosphate-labeled HEK293 cells co-expressing HA-tagged WT or D478A kinase-inactive (KI) Fam20C with the indicated substrates (n = 3). (B) Residues of Calsequestrin 2 (CSQ2) corresponding to the only Fam20C S-x-E phosphorylation motif on the protein are highlighted in blue. (C) Representative autoradiograph and Flag IB of Flag IPs of [32P]-orthophosphate-labeled rat cardiomyoblast H9C2 cells expressing the indicated variant of CSQ2 (n = 3). (D) Representative autoradiograph of time-dependent incorporation of $[\gamma\text{-}^{32}\text{P}]$ ATP into CSQ2 using purified proteins in an in vitro kinase assay (n = 3). CSQ2-His was expressed and purified from *E. coli* and Fam20C WT or KI was purified from baculovirus. Corresponding IB of α-His is shown below. (E) Turbidity of un- and phosphorylated-CSQ2 as measured by absorbance at 350 nm (n = 3). (F) Model demonstrating the consequence of Fam20C phosphorylation of CSQ2.
DOI: https://doi.org/10.7554/eLife.41378.002

determined to be phosphorylated by WT Fam20C. It is likely that these phosphorylations will have multifaceted contributions to ER/SR $Ca^{2+}$ regulation (see discussion). We note that triadin (and Stim-1, below) is known to be phosphorylated on its cytosolic portion, explaining the background in signal in the Fam20C KI lane. Next, we chose to focus on CSQ2 (*Houle et al., 2004*) and Stim1 (*Collins et al., 2013*) given the significance of their physiological roles.

CSQ2 is the primary $Ca^{2+}$ binding protein in the SR and is important for SR $Ca^{2+}$ regulation and cardiac pathophysiology (*Györke and Terentyev, 2008*; *Faggioni and Knollmann, 2012*). It is intriguing that CSQ2 has a single, conserved S-x-E site (Ser385) on its C-terminal tail (*Figure 1B*), the region of the protein necessary for proper $Ca^{2+}$ binding (*Shin et al., 2000*) and polymerization (*Park et al., 2003*). Transfection of Flag-tagged WT and Ser385Ala mutant CSQ2 constructs into

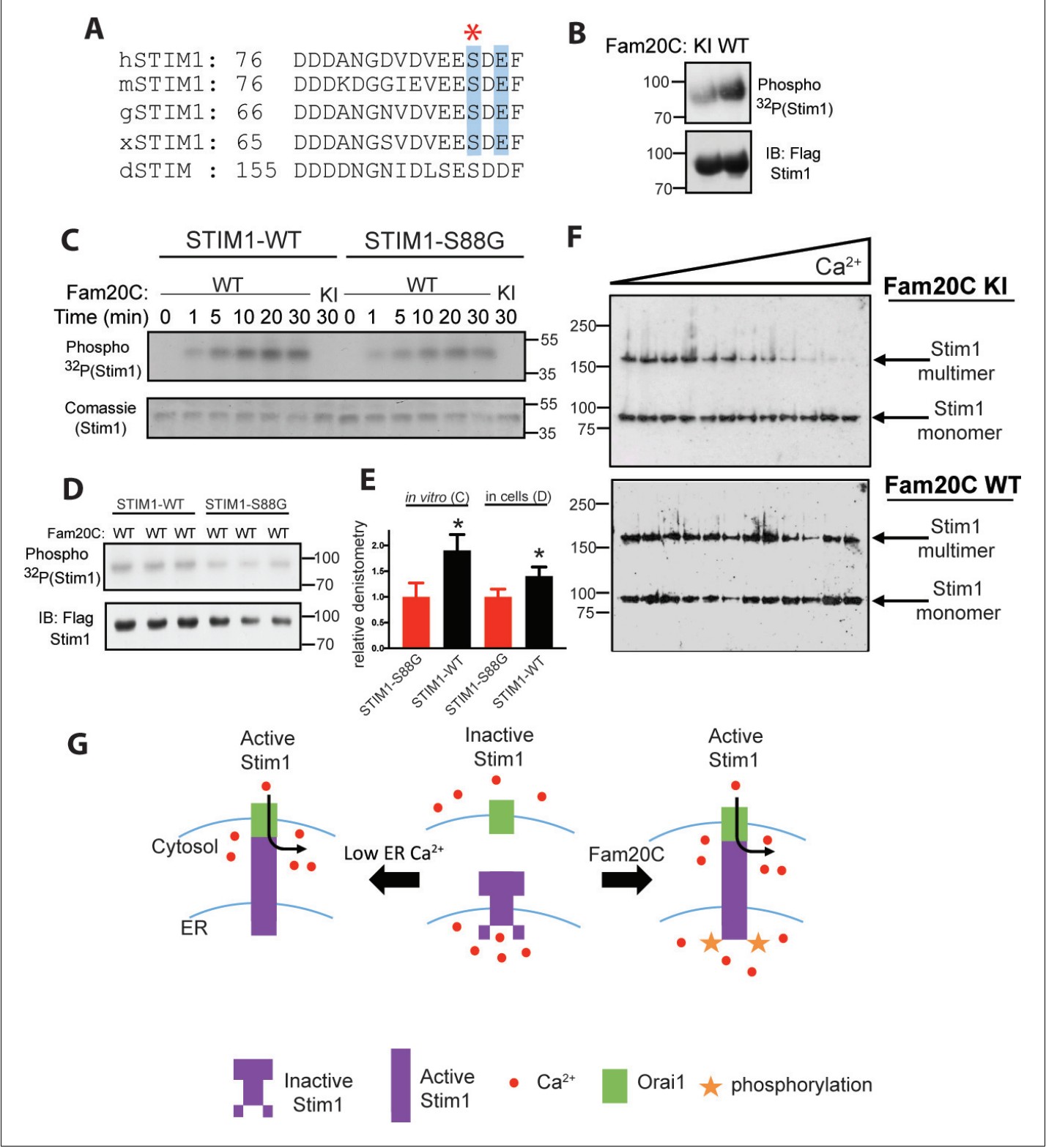

**Figure 2.** Fam20C phosphorylation of Stim1 regulates Stim1 activation (**A**) Residues of Stim1 from the canonical Ca²⁺-binding EF-hand loop corresponding to the Fam20C S-x-E phosphorylation motif are highlighted in blue.  Heart disease associated Stim1-Ser88 is noted with an asterisk (*h*: human, *m*: mouse, *g*: gallus, *x*: xenopus, *d*: drosophila). (**B**) Representative autoradiograph and Flag IB of Flag IPs of ³²P-orthophosphate-labeled HEK293 cells co-expressing Stim1-Flag and HA-tagged WT or D478A kinase-inactive (KI) Fam20C (n = 3). (**C**) Representative autoradiograph of time-dependent incorporation of [γ-³²P] ATP into Stim one using purified proteins in an in vitro kinase assay. His-tagged Stim1 was expressed and purified

*Figure 2 continued on next page*

Figure 2 continued

from *E. coli,* and Fam20C WT or KI was purified from baculovirus (n = 6). Corresponding coomassie staining is shown below. (D) Representative autoradiograph and Flag IB of Flag IPs of $^{32}$P-orthophosphate-labeled HEK293 cells co-expressed with Stim1 and WT Fam20C (n = 6). (E) Densitometry relative to protein control of autoradiograph from C and D (at 30 min time point). (F) Representative western blots showing the crosslinking of Stim1 (A230C) co-expressed with either Fam20C KI (top), or Fam20C WT (bottom), in cellular membranes incubated with increasing Ca$^{2+}$ concentrations (0–2.0 mM) (n = 3). Crosslinked, multimerized (active) Stim one is indicated by the upper arrow, and monomeric (inactive) Stim1 is indicated by the lower arrow. (G) Model demonstrating Stim1 activation via either low ER Ca$^{2+}$ or Fam20C phosphorylation. Data are represented as the mean ±SEM. *p < 0.05 by Student's *t* test.

DOI: https://doi.org/10.7554/eLife.41378.003

H9C2 rat myoblast cardiac cells, followed by $^{32}$P-orthophosphate labeling, revealed that only WT CSQ2 was phosphorylated (*Figure 1C*), suggesting that CSQ2-Ser385 is phosphorylated. We then purified recombinant His-tagged CSQ2 from *E. coli* and performed an in vitro kinase assay with purified recombinant Fam20C and $^{32}$P-labeled ATP (*Figure 1D*). Mutation of CSQ2 Ser385 to Ala abrogated phosphorylation, but mutation of Ser393 to Ala did not, suggesting that Fam20C phosphorylates CSQ2 only at Ser385.

We sought to determine if Fam20C phosphorylation of CSQ2 affected CSQ2's ability to polymerize, which is critical to its functional roles (*Faggioni and Knollmann, 2012*). To study this, we employed a turbidity assay (*Sanchez et al., 2011*), where absorbance at 350 nM was monitored as a function of Ca$^{2+}$, using CSQ2 that was either unphosphorylated or phosphorylated in vitro by Fam20C (*Figure 1E*). Indeed, we found that phosphorylated CSQ2 showed a greater propensity to form higher order polymeric structures than unphosphorylated CSQ2 (*Figure 1E and F*).

Stim1 is an ER/SR luminal Ca$^{2+}$ sensor that regulates store-operated Ca$^{2+}$ entry (SOCE) and other Ca$^{2+}$ sensing mechanisms in a variety of cell types (*Soboloff et al., 2012*; *Bénard et al., 2016*), including cardiomyocytes, where it plays a role in preventing heart failure following induced pressure overload or aging (*Bénard et al., 2016*). We identified a conserved S-x-E site in the Ca$^{2+}$-binding EF-hand region of the luminal portion of Stim1 (*Figure 2A*) (*Stathopulos et al., 2008*). Stim1 was shown to be phosphorylated by Fam20C in cells and in vitro (*Figure 2B and C*) using the same techniques as above (*Figure 1*). In addition, Stim1-Ser88Gly (*Figure 2A*), a mutation recently discovered in a patient with heart disease (*Harris et al., 2017*) (see discussion), is less robustly phosphorylated than Stim1-WT in vitro and in cells (*Figure 2C,D and E*), suggesting that Fam20C phosphorylates Stim1-Ser88. Next, we employed a recently developed approach to measure Stim1 activity in ER membranes that relies on an engineered cysteine in Stim1 (A230C) to capture the conformational change involving the interaction between paired luminal domains that accompanies its activation (*Hirve et al., 2018*). Stim1 activation involves a dramatic reorganization of the protein complex, where the transmembrane regions of two Stim1 molecules (where the engineered cysteine are) align close together, allowing the complex to extend and activate plasma membrane Ca$^{2+}$ channels (*Hirve et al., 2018*). We determined that over-expression of WT Fam20C in cells caused significantly increased Stim1 activation under physiological Ca$^{2+}$ levels in comparison to the catalytically inactive KI Fam20C (*Figure 2F and G*). Taken together, phosphorylation of Ca$^{2+}$ handling proteins in the ER/SR by Fam20C is likely playing an important and complex regulatory role.

## Fam20C regulates SR Ca$^{2+}$ handling in isolated myocytes

Given that Fam20C phosphorylates proteins that reside in the lumen of the SR (*Figures 1* and *2*), we hypothesized that Fam20C plays a role in cardiomyocyte SR Ca$^{2+}$ handling. To determine this, we developed a mouse model with LoxP sites engineered between exons 6 and 10 of *Fam20c* (*Fam20c$^{Fl/FL}$*) (*Figure 3—figure supplement 1*). We crossed these mice with α-MHC-Cre mice to knockout Fam20C in ventricular myocytes, generating *Fam20c$^{Fl/FL}$*α-MHC-Cre-positive (Fam20C cKO) mice and *Fam20c$^{Fl/FL}$*α-MHC-Cre-negative (Fam20C WT) littermates. Importantly, we previously showed that this α-MHC-Cre mouse line has normal cardiac function following aging (*Fang et al., 2017*).

We examined intracellular Ca$^{2+}$ transients and cell shortening of isolated cardiomyocytes from Fam20C cKO and WT mice paced at 0.5 Hz field stimulation (*Figure 3A and B*). We analyzed both baseline (2 months old) and 9 months old aged mice, which can be more predisposed to compromised cardiac function (*Bénard et al., 2016*). A decrease in the peak amplitude (*Figure 3C*),

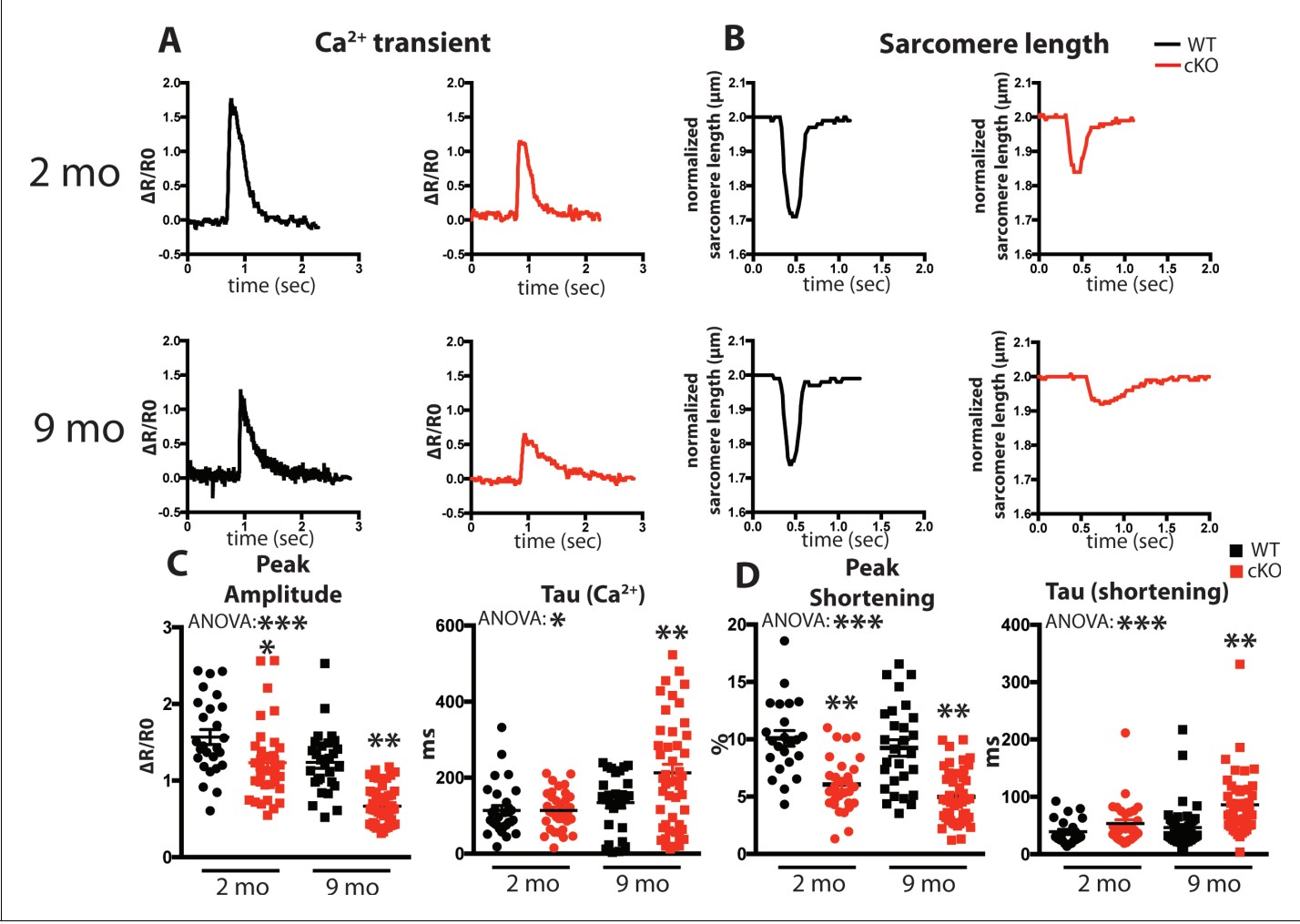

**Figure 3.** Fam20C regulates $Ca^{2+}$ handling in isolated cardiomyocytes. (A and B) Tracing of 0.5 Hz stimulated isolated cardiomyocytes for (A) $Ca^{2+}$ transients (change between the basal and peak ratio (ΔR) of Fura2-AM fluorescence) and (B) sarcomere length for 2 (top) and 9 (bottom) months old mice. (C) Quantitation of normalized peak amplitude (left) and relaxation constant tau (right) of $Ca^{2+}$ transients from A. (D) Quantitation of peak amplitude changed (left) and relaxation constant tau (right) of sarcomere length from B (n = 3 mice, 15–25 cardiomyocytes per mice). Data are represented as the mean ±SEM. *p < 0.05; **p < 0.01, by Student's *t* test or ANOVA (when indicated) for WT v. cKO.

DOI: https://doi.org/10.7554/eLife.41378.004

The following figure supplement is available for figure 3:

**Figure supplement 1.** Generation of cardiac specific Fam20C cKO mice.

DOI: https://doi.org/10.7554/eLife.41378.005

corresponding to the extent of SR $Ca^{2+}$ release, and an increase in the relaxation constant tau (*Figure 3C*), corresponding to the time it takes for SR $Ca^{2+}$ reuptake, was observed for the $Ca^{2+}$ transients of 9 months old Fam20C deficient cardiomyocytes in comparison to controls. Similar results were obtained for the cell shortening measurements (*Figure 3D*). We note mildly reduced cardiomyocyte function at baseline. Strikingly, following aging-induced stress we identified severe cardiomyocyte defects from Fam20C deficiency, particularly in cardiomyocyte relaxation, which is likely causative of aging-induced diastolic heart failure (see below). These results establish Fam20C as an important regulator of SR $Ca^{2+}$ cycling in stressed cardiomyocytes.

## Fam20C regulates cardiac relaxation and contractility

Based on the cardiomyocyte $Ca^{2+}$ cycling defects observed from Fam20C cKO mice upon aging (*Figure 3*), we sought to determine the role of Fam20C in contractility and relaxation in vivo via invasive

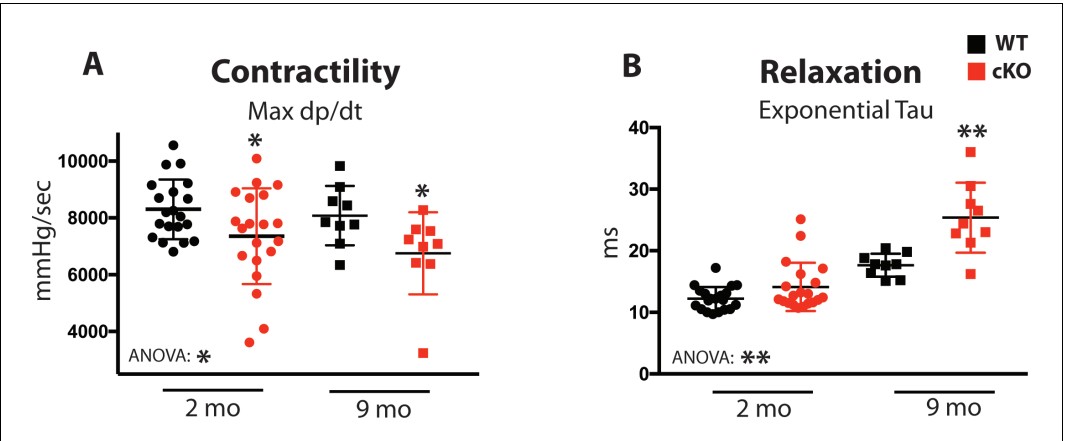

**Figure 4.** Fam20C regulates cardiac contractility and relaxation in vivo. Hemodynamic assessment of contraction and relaxation (n = 9–20). (**A**) Assessment of maximum pressure difference over time (d*P*/d*t*). (**B**) Assessment of the isovolumetric relaxation constant tau. Data are represented as the mean ±SEM. *p < 0.05; **p < 0.01, by Student's *t* test or ANOVA (when indicated) for WT v. cKO.

DOI: https://doi.org/10.7554/eLife.41378.006

The following figure supplement is available for figure 4:

**Figure supplement 1.** Role of Fam20C in the acute hemodynamic response to β-adrenergic receptor activation.

DOI: https://doi.org/10.7554/eLife.41378.007

hemodynamic measurement of LV pressure. Contractility and relaxation are controlled by SR $Ca^{2+}$ release and reuptake, respectively. At 2 months of age, we observe mild but statistically significant defects of cardiac contractility (Max d*P*/d*t*) (*Figure 4A*) and relaxation (Min d*P*/d*t*) (*Figure 4—figure supplement 1A*) in Fam20C cKO mice in comparison to WT mice, but no differences to the heart rate and other hemodynamic parameters were observed (*Figure 4—figure supplement 1B-D*).

Upon 9 months of aging, relaxation was significantly blunted in Fam20C cKO mice (*Figure 4—figure supplement 1A*). The isovolumetric relaxation constant tau was dramatically higher (*Figure 4B*), demonstrating delayed cardiac relaxation, and indicating overall diastolic dysfunction for 9 months old Fam20C cKO mice, which is likely to cause heart failure (see below). Importantly, these results are consistent with the in vitro data (*Figure 3*). Interestingly, we found that Fam20C's role in contractility and relaxation is independent of acute β-adrenergic signaling in 2 months old mice, via infusion of either dobutamine or esmolol (*Schmid et al., 2015*) (*Figure 4—figure supplement 1*).

## Fam20C cKO mice develop heart failure, fibrosis, apoptosis, and DCM following aging

Due to the defects observed in SR $Ca^{2+}$ cycling (*Figure 3*) and cardiac relaxation and contractility (*Figure 4*), we sought to determine Fam20C's temporal role in cardiac function in vivo by performing serial 2D echocardiography on mice who were from 2 to 9 months of age. Two and six months old Fam20C cKO mice were indistinguishable from their WT littermates. However, following 9 months of aging, Fam20C cKO mice displayed a decided increase in LV chamber size in both diastole and systole (*Figure 5A*). LV systolic function, as measured by fractional shortening (% FS), was dramatically decreased in Fam20C cKO mice (*Figure 5A*), suggesting that these mice were developing heart failure. Depressed systolic function was accompanied by a thinning of the LV walls (*Figure 5B*). Furthermore, measurement by qRT-PCR on 9 months old Fam20C cKO mice showed significant increases in the re-expression of fetal gene markers of cardiac hypertrophy and heart failure, including atrial natriuretic factor (ANP), brain natriuretic peptide (BNP), and β-myosin heavy chain (β-MHC), in comparison to controls (*Figure 5C*). Gross morphological analysis showed enlarged hearts (*Figure 6A*), consistent with the development of DCM following 9 months of aging. The observed heart failure in the Fam20C cKO mice following aging is likely due to dramatic SR $Ca^{2+}$ handling defects and delayed cardiac relaxation (*Figures 3* and *4*).

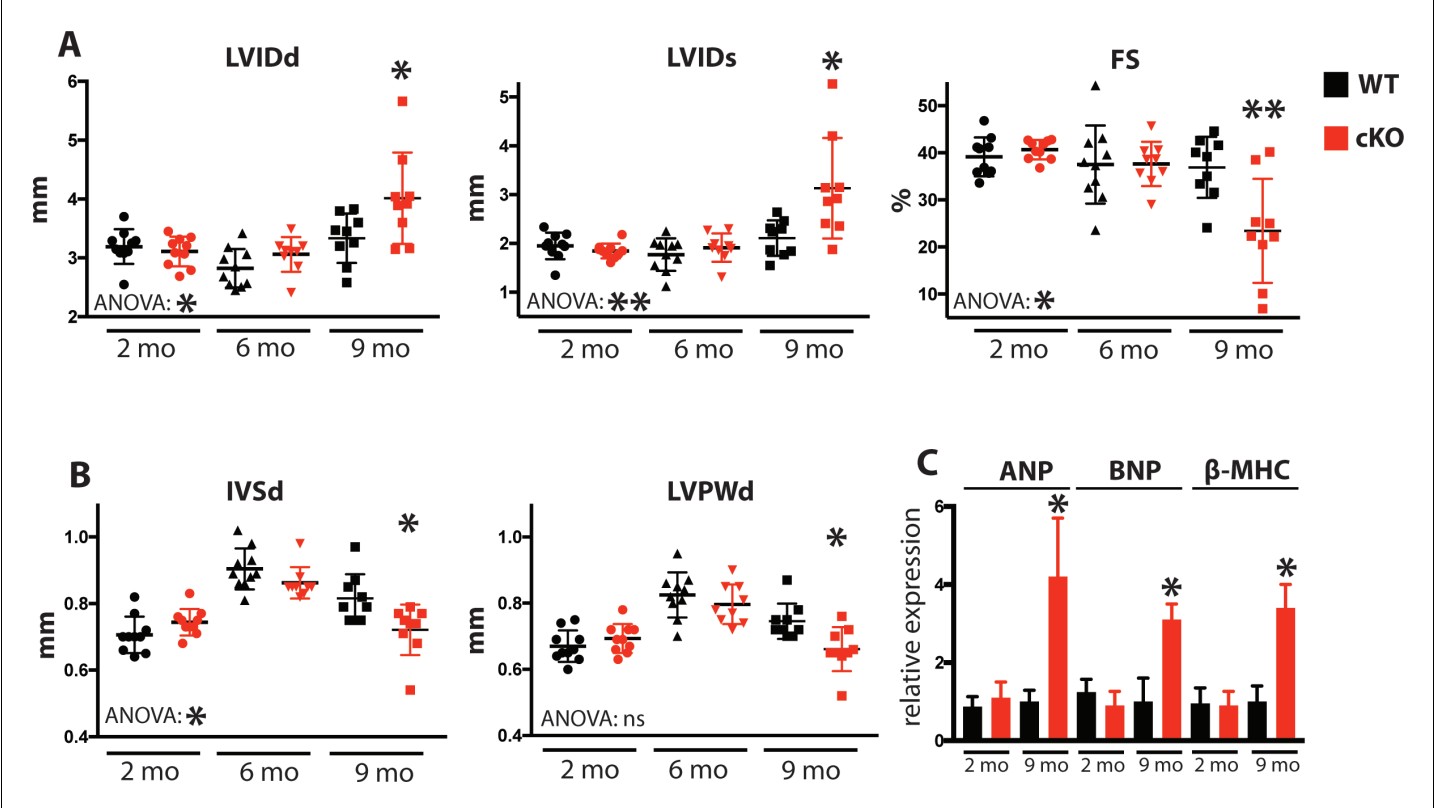

**Figure 5.** Fam20C cKO mice develop heart failure upon aging. (**A and B**) Echocardiographic measurements for *Fam20C*^Fl/Fl α-MHC-Cre-positive (cKO) and α-MHC-Cre-negative littermate control mice (WT) (n = 10 mice at 2, 6, and 9 months) of (**A**) LVIDd, LVIDs, LV FS, (**B**) IVSd, and LVPWd. (**C**) RT-qPCR of cardiac fetal gene markers of Fam20C WT and cKO mice at 2 and 9 months (n = 3). Data are represented as the mean ±SEM. *p < 0.05; **p < 0.01, or ns (not significant) by Student's *t* test or ANOVA (when indicated) for WT v. cKO.
DOI: https://doi.org/10.7554/eLife.41378.008

Mouse hearts were dissected, fixed, and subjected to histological analysis by microscopy. In 2 months old Fam20C WT or cKO mice, no abnormalities or differences were identified. However, following 9 months of aging in the Fam20C cKO mice, Hematoxylin-Eosin (H and E) stained cardiac sections demonstrated ventricular dilation (*Figure 6B*), consistent with the echocardiographic analysis (*Figure 5A*).

Cardiac fibrosis and apoptosis strongly associate with heart failure (*Reed et al., 2011*; *Segura et al., 2014*). Using Masson's Trichrome staining, we established that interstitial fibrosis was greatly increased in Fam20C cKO mice, while WT mice demonstrated no signs of fibrosis following 9 months of aging (*Figure 6C*). Gravimetric analysis showed that the heart weight to body weight ratio was greater in Fam20C cKO mice (*Figure 6D*). Terminal deoxynucleotidyltransferase-mediated dUTP nick end labeling (TUNEL) staining demonstrated significantly more apoptotic nuclei in Fam20C cKO mice (*Figure 6D* and *Figure 6—figure supplement 1A*). The amount of cleaved poly (ADP-ribose) polymerase (PARP), a marker of apoptosis, was increased in Fam20C cKO heart lysates as determined by immunoblotting (*Figure 6—figure supplement 1B*). Finally, profibrotic gene expression markers: α-smooth muscle actin (α-SMA), procollagen types I α1 (Coll-1a1) and III α1 (Coll-3a1), and connective tissue growth factor (CTGF) were increased in Fam20C cKO mice upon aging (*Figure 6—figure supplement 1C*).

## Induced pressure overload causes accelerated heart failure in Fam20C deficient mice

To determine Fam20C's role in heart function during sustained pathological stress in vivo, we employed the transverse aortic constriction (TAC)-induced pressure overload model (*Figure 7*). The

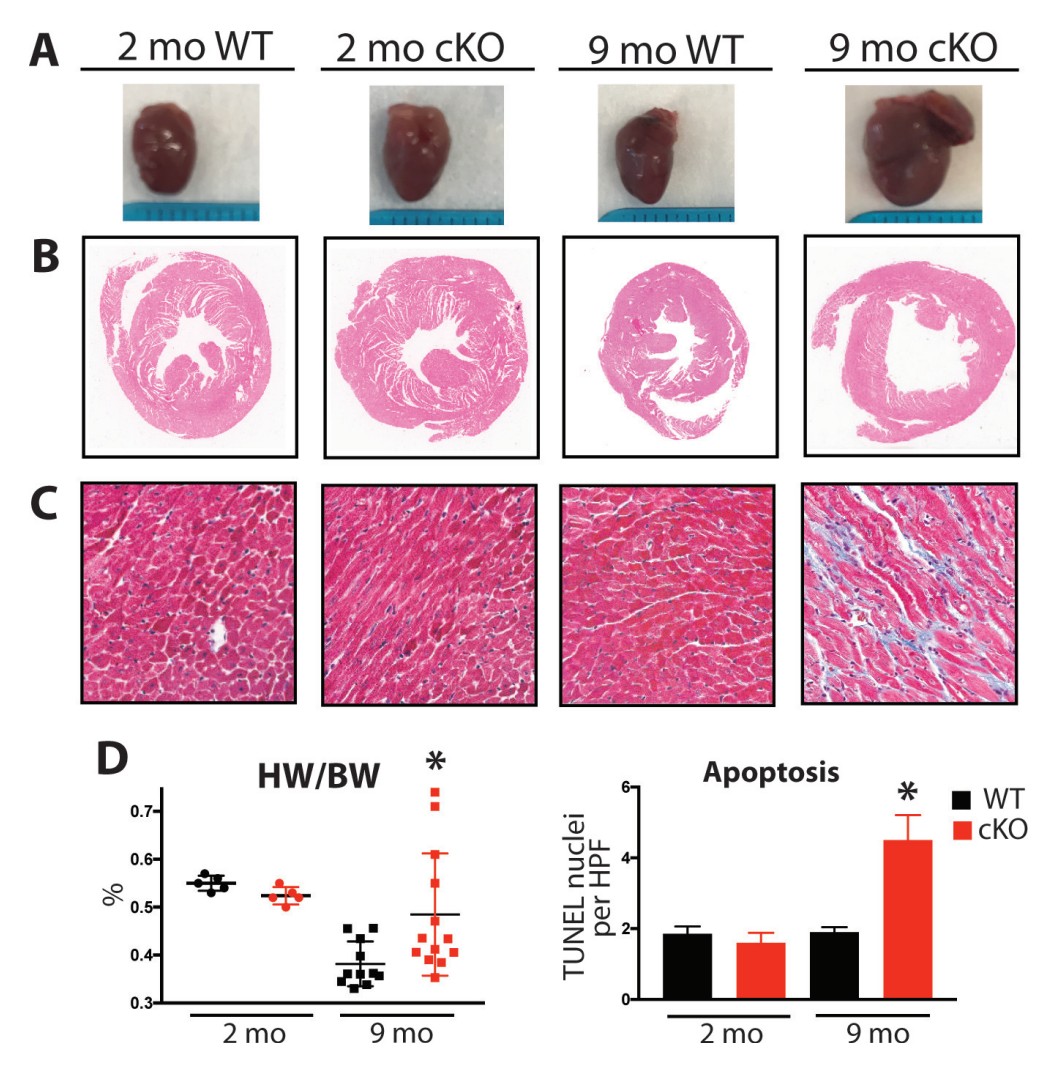

**Figure 6.** Aging Fam20C cKO mice develop fibrosis, apoptosis, and DCM. (A) Representative macroscopic view of whole mouse hearts. (B and C) Representative microscopic view of (B) H and E stained (1x magnification) and (C) Masson's Trichrome stained (40x magnification) cardiac sections (n = 3). (D) Heart weight to body weight ratios (n = 5–13) (left). Quantitation of TUNEL positive nuclei per high powered field (HPF) (n = 3 mice) (right). Data are represented as the mean ±SEM. *p < 0.05; **p < 0.01, by Student's t test for WT v. cKO.

DOI: https://doi.org/10.7554/eLife.41378.009

The following figure supplement is available for figure 6:

**Figure supplement 1.** Aging Fam20C cKO mice develop apoptosis and fibrosis.

DOI: https://doi.org/10.7554/eLife.41378.010

TAC model, which causes hypertrophy and eventual heart failure (*Rockman et al., 1991*), is a more targeted and immediately severe stressor than aging, and can directly test Fam20C's role in the hypertrophic response. The response to pressure overload for 4 weeks was similar between genotypes (*Figure 7—figure supplement 1A*) of 2 months old Fam20C WT and cKO mice, both of which developed LV wall thickening (*Figure 7—figure supplement 1B and C*). However, Fam20C cKO mice developed significantly exaggerated LV chamber dilation (*Figure 7A*) and showed further deterioration in cardiac function (*Figure 7A*) in comparison to their WT littermates. Gravimetric analysis of isolated hearts showed that the heart weight to body weight ratio was higher in Fam20C cKO mice (*Figure 7B*). Re-expression of cardiac fetal genes (*Figure 7B*) further confirmed the presence of exaggerated heart failure in Fam20C cKO mice in comparison to WT littermates. We determined

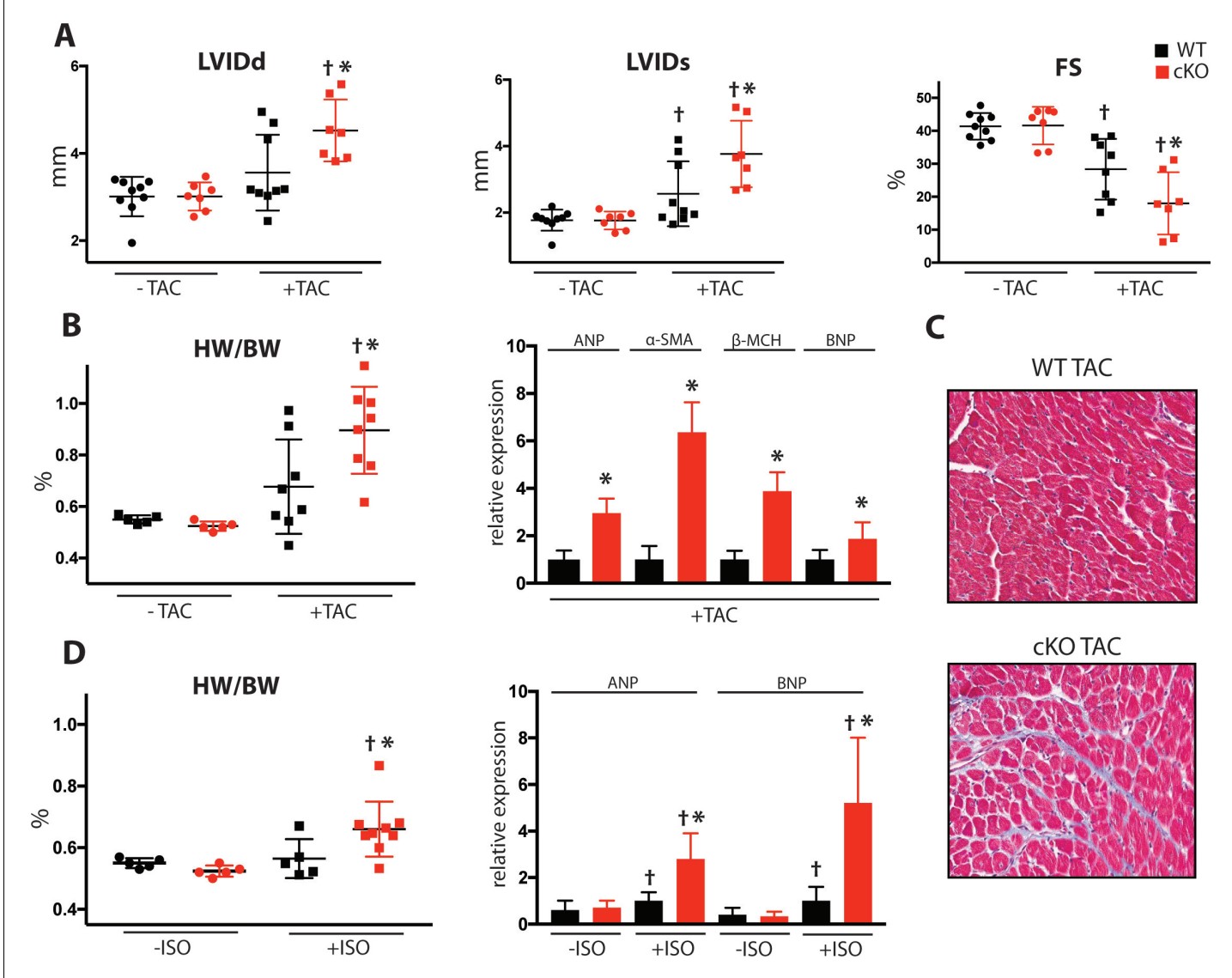

**Figure 7.** Fam20C cKO mice develop heart failure following induced pressure overload. (A) Echocardiographic measurements for baseline (2 months old) mice and after 4 weeks of TAC (n = 7–9) of LVIDd, LVIDs, and LV FS. (B) Heart weight to body weight ratios (n = 5–9) (left). RT-qPCR of cardiac fetal gene markers following TAC (n = 3) (right). (C) Representative microscopic view of Masson's Trichrome stained (40x magnification) cardiac sections following TAC (n = 3 mice). (D) Mice were subjected to 2 weeks of chronic Isoproterenol (ISO) infusion (n = 5–9). Heart weight over body weight ratio (n = 5–9) (left) and RT-qPCR of cardiac fetal gene markers (n = 3) (right). Data are represented as the mean ±SEM. *p < 0.05, **p < 0.01, for WT v. cKO; † p < 0.05 for TAC or ISO treatment by Student's t test.

DOI: https://doi.org/10.7554/eLife.41378.011

The following figure supplement is available for figure 7:

**Figure supplement 1.** Induced pressure overload causes hypertrophy, apoptosis in Fam20C cKO mice.

DOI: https://doi.org/10.7554/eLife.41378.012

also that Fam20C cKO mice heart sections showed increased fibrosis (*Figure 7C*) and apoptosis (*Figure 7—figure supplement 1D and E*) following TAC.

To further evaluate Fam20C's role in stress-induced cardiac remodeling, we administered chronic β-adrenergic stimulation by isoproterenol infusion to Fam20C cKO and WT mice using mini-osmotic pumps (*Thum et al., 2008*). After 2 weeks, Fam20C cKO mice showed a significantly increased heart weight to body weight ratio (*Figure 7D*), and increased fetal gene re-expression (*Figure 7D*) in comparison to controls. Taken together, both TAC and isoproterenol infusion caused Fam20C cKO mice

to develop an exaggerated depression of LV function in comparison to WT mice, reminiscent of the aging model seen above.

## Fam20C regulates ER $Ca^{2+}$ homeostasis

Given that Fam20C is widely expressed in mammals (*Nalbant et al., 2005*) and that ER/SR $Ca^{2+}$ homeostasis is important for many cell types, we sought to determine Fam20C's role in cellular $Ca^{2+}$ homeostasis in non-muscle cells. We transfected HeLa cells with either Flag-tagged WT or KI Fam20C and tested for constitutive $Ca^{2+}$ influx by briefly changing the external solution to 0 mM $Ca^{2+}$ and then back to 2 mM $Ca^{2+}$ (*Figure 8A*), as previously described (*Hirve et al., 2018*). Only the WT Fam20C transfected cells showed a significant increase over control HeLa cells in $Ca^{2+}$ influx, prior to thapsigargin stimulation which causes comparable $Ca^{2+}$ influx for each sample. We note that Fam20C mediated activation of Stim1 has an effect similar to Stim1 activating mutations (*Hirve et al., 2018*). This indicates that Fam20C regulates store-operated $Ca^{2+}$ entry, likely via activation of Stim1 (*Figure 2E and F*), but possibly through the phosphorylation of other ER substrates. Moreover, this demonstrates that Fam20C plays a role in regulating cellular ER $Ca^{2+}$ signaling.

Using an alternative approach to study Fam20C's role in ER $Ca^{2+}$ homeostasis, we treated cells with thapsigargin, a SERCA inhibitor, which results in the release of ER $Ca^{2+}$ stores (*Wiley et al., 2013*). In WT and Fam20C shRNA knockdown (shFam20C) U2OS cell lines (*Tagliabracci et al., 2015*) we observed that thapsigargin-induced release of ER $Ca^{2+}$ was reduced upon Fam20C knockdown (*Figure 8B*), showing Fam20C's role in maintaining ER $Ca^{2+}$ storage. Because dysregulation of

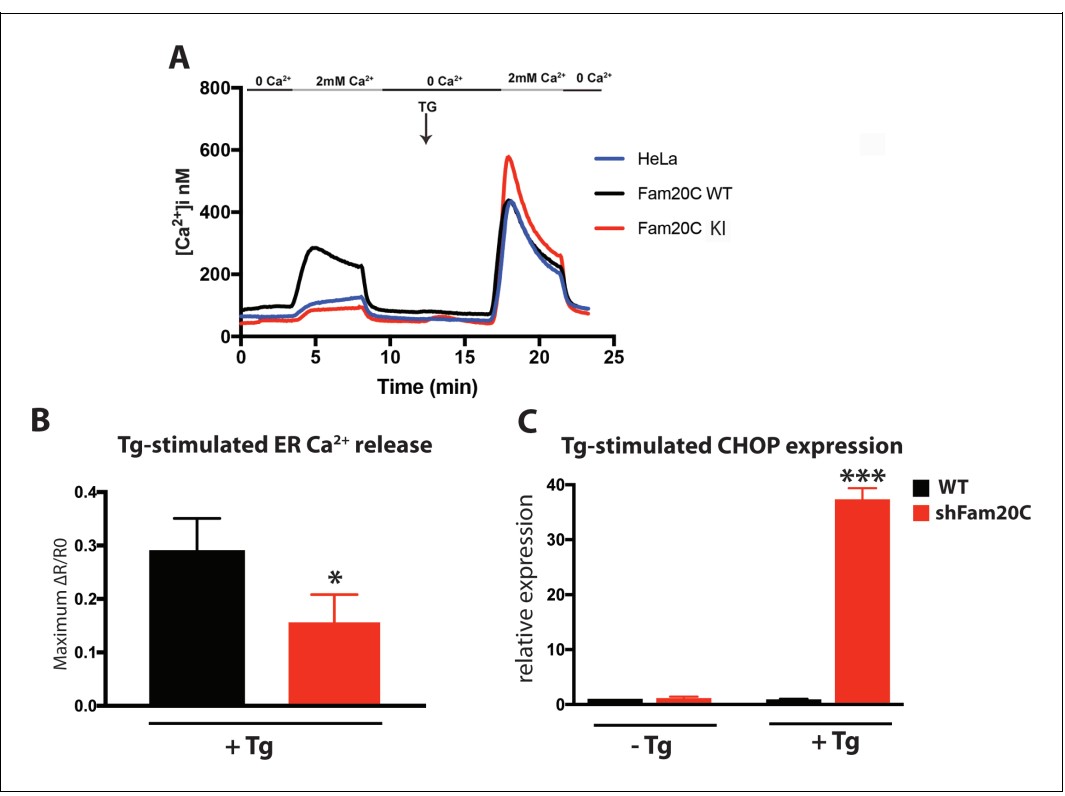

**Figure 8.** Fam20C regulates ER $Ca^{2+}$ homeostasis. (**A**) Single-cell $[Ca^{2+}]i$ measurements in HeLa cells expressing Fam20C WT (n = 156), Fam20C KI (n = 139), or in non-transfected HeLa cells (HeLa; n = 126). Cells were exposed to solutions containing varied concentrations of $CaCl_2$ or 1 µM thapsigargin (TG) as indicated. (**B**) Peak $Ca^{2+}$ release (maximum minus basal, peak ratio (ΔR) of Fura2-AM fluorescence) of WT and Fam20C shRNA knockdown (shFam20C) U2OS cells following 2 µM thapsigargin (Tg) treatment. (**C**) RT-qPCR of UPR response gene CHOP expression of WT and shRNA Fam20C knockdown U2OS cells following mild stimulation with thapsigargin (250 nM, 4 hr). Data are represented as the mean ±SEM. *p < 0.05; ***p < 0.001, by Student's *t* test for WT v. shFam20C.

DOI: https://doi.org/10.7554/eLife.41378.013

ER $Ca^{2+}$ induces the unfolded protein response (UPR) in cells, we assessed WT and shFam20C U2OS cell lines for UPR induction using qRT-PCR to monitor levels of C/EBP homologous protein (CHOP), a common UPR marker (*Wiley et al., 2013*) (*Figure 8C*). There was no basal UPR detected; however, shFam20C U2OS cells showed a robust expression of CHOP following stimulation with a mild amount of thapsigargin, at concentrations that did not elicit a response in WT cells. This suggests that the shFam20C cells were already compromised in their ER $Ca^{2+}$ homeostasis and were predisposed to $Ca^{2+}$ stress-induced induction of the UPR. Together, these data (*Figure 8*) suggest a general role for Fam20C in regulating ER $Ca^{2+}$ homeostasis.

## Discussion

We have determined the biochemical and physiological function of a kinase (Fam20C) that phosphorylates proteins within the lumen of the cardiac SR. Fam20C is the primary protein kinase within a small family of putative kinases, all of which have a signal peptide that guides them to the lumen of the ER/SR (*Tagliabracci et al., 2013*; *Tagliabracci et al., 2015*). Interest in Fam20C's role in heart function came from our observation that Fam20C phosphorylates HRC on Ser96 (*Pollak et al., 2017*). DCM patients with the common HRC-Ser96Ala polymorphism develop deadly arrhythmias (*Arvanitis et al., 2008*), likely due to loss of Fam20C phosphorylation at that site. The clinical significance of Fam20C phosphorylation is highlighted by the fact that this polymorphism is predicted to be present in 60% of the population (*Tzimas et al., 2017*), and that roughly one in 2,500 Americans has DCM (*Benjamin et al., 2018*). A comparison, however, of our results with prior studies suggests that Fam20C's physiological roles extend beyond phosphorylation of HRC-Ser96. For example, HRC-Ser96Ala knock-in mice were only mildly deficient in cardiac function following aging in comparison to HRC-Ser96 mice (*Singh et al., 2013*; *Tzimas et al., 2017*).

In addition to HRC, here we have identified multiple novel Fam20C substrates that reside in the SR (*Figures 1* and *2*). While our data indicate that these phosphorylations affect SR $Ca^{2+}$ homeostasis, dissecting the individual contribution of each substrate is likely to be very complex and remains to be determined. The biochemical functions of Fam20C phosphorylation remain mostly unknown but have been demonstrated in some instances. For example, Fam20C mediated phosphorylation of HRC regulates its binding to SERCA2a and triadin, the latter of which is important for preventing arrhythmias (*Pollak et al., 2017*).

Here we show that Fam20C phosphorylation of CSQ2 increases its ability to polymerize (*Figure 1*). This result is consistent with a previous report using phospho-mimetics of CSQ2 (*Sanchez et al., 2011*). CSQ2 is expressed in cardiac muscle, while the closely related calsequestrin 1 (CSQ1) regulates SR function in skeletal muscle (*Sanchez et al., 2012*). It is intriguing that CSQ2 has a disordered, acidic C-terminal tail which contains an S-x-E site, but CSQ1 does not. This suggests that CSQ2 is differentiated from CSQ1 via regulation by Fam20C phosphorylation. In response to increasing amounts of $Ca^{2+}$, CSQ2 transitions from a monomer to higher-ordered polymeric states (*Figure 1F*) (*Györke and Terentyev, 2008*). As a monomer, CSQ2 binds more effectively to triadin; upon polymerization, it more effectively sequesters $Ca^{2+}$. Therefore, Fam20C phosphorylation likely effects CSQ2's $Ca^{2+}$ binding and its ability to bind to and regulate triadin. Interestingly, the physiologically critical KEKE motif of triadin (*Figure 1A*) contains an S-x-E site (*Terentyev et al., 2005*; *Kobayashi et al., 2000*), whose phosphorylation status may affect triadin's ability to bind to its protein partners: HRC, CSQ2, and RyR2. Finally, sarcalumenin, calumenin, and calreticulin regulate cardiac function through their interactions with SERCA2a; we anticipate that their phosphorylation may alter their interactions with SERCA2a, similarly to HRC (*Pollak et al., 2017*).

We also show that Fam20C mediated phosphorylation of Stim1 shifts the $Ca^{2+}$ concentration dependence of the Stim1 conformational change and thereby favors Stim1 activation (*Hirve et al., 2018*) (*Figure 2*). Given the location of the phosphorylation on the EF-hand motif, it is likely that it acts to modulate Stim1 $Ca^{2+}$ binding, which is the basis of its activity as an ER/SR $Ca^{2+}$ sensor (*Soboloff et al., 2012*); alternatively, Fam20C phosphorylation may induce a structural change favoring Stim1 activation. Interestingly, a recently identified mutation at the phosphorylation site (Stim1-Ser88Gly) has been linked to cardiac dysfunction in patients; importantly, this mutation causes $Ca^{2+}$ dysregulation in cells (*Harris et al., 2017*). This patient is one of three patients identified to date where a mutation to Stim1's luminal EF-hand causes cardiac dysfunction (*Harris et al., 2017*;

*Walter et al., 2015*). However, further experimentation is necessary to directly show Fam20C mediated phosphorylation of Stim1 regulates cardiomyocyte SOCE.

It is intriguing that Stim1 is widely expressed in mammals, and therefore when mutated is involved in multiple pathological roles (*Soboloff et al., 2012*), further suggesting a broad regulatory function for Fam20C in ER/SR Ca$^{2+}$ regulation (*Figure 8*) that is likely to be a general mechanism in all cells, which can be addressed by future studies. This is an extension of the previously appreciated role of Fam20C in the phosphorylation of only secreted proteins, and this extension is in-line with a recent report showing Fam20's role in regulating ER redox homeostasis (*Zhang et al., 2018*).

Physiologically, we show that loss of murine cardiomyocyte *Fam20c* causes accelerated deterioration of cardiac function following various pathological stimuli, including aging, TAC, and isoproterenol infusion (*Figures 5–7*). These severe functional consequences are likely due to loss of SR luminal protein phosphorylation. The mild alterations in cardiomyocyte performance (*Figure 3*) and pressure development (*Figure 4*) at baseline (2 months old) – where no functional abnormalities were observed (*Figure 5*) – foreshadow the significant defects found in cardiomyocyte function and cardiac contractility and relaxation following aging which accompanies heart failure. Fam20C has not been previously demonstrated to phosphorylate SR Ca$^{2+}$ handling proteins or play a critical role in stress-induced heart failure development. There is broad precedent for this general concept, however, as CaMKII mediated RyR2 phosphorylation (*Ling et al., 2009*) and PKA mediated phospholamban phosphorylation (*Braz et al., 2004*) have established roles in the development of heart failure.

In summary, we present protein phosphorylation by Fam20C within the lumen of the ER/SR as an important regulatory mechanism that controls cellular Ca$^{2+}$ homeostasis and cardiac pathophysiology. Most strikingly, aging led to significantly worsened systolic and diastolic function, associated with interstitial fibrosis, in Fam20C cKO mice. Interestingly, in adult patients, approximately 50% of those suffering from congestive heart failure display a preponderance of diastolic dysfunction, often associated with myocardial fibrosis as they age (*Ouzounian et al., 2008*). By establishing Fam20C as a regulator during aging of myocardial form and function, this study portends an important molecular biologic role for Fam20C in the modern epidemic of congestive heart failure.

# Materials and methods

## Key resources table

| Reagent type (species) or resource | Designation | Source or reference | Identifiers | Additional information |
|---|---|---|---|---|
| Antibody | Mouse monoclonal M2 FLAG | Sigma | A2220-5 ml, RRID:AB_439685 | (1:100) |
| Antibody | Rabbit polyclonal anti-Flag | Sigma | F7425-.2MG, RRID:AB_796202 | (1:1000) |
| Antibody | Rabbit polyclonal anti-Stim1 | Cell Signaling | 4916S, RRID:AB_1849882 | (1:1000) |
| Antibody | Rabbit polyclonal anti 6x-His | Thermo Fisher | MA1-21315-HRP, RRID:AB_10977997 | (1:2000) |
| Antibody | Rabbit polyclonal anti Bip/GRP78 | Abcam | ab21685, RRID:AB_880312 | (1:1000) |
| Antibody | Rabbit polyclonal anti Calsequestrin 2 | Thermo Fisher | PA1-913, RRID:AB_2540244 | (1:5000) |

*Continued on next page*

*Continued*

| Reagent type (species) or resource | Designation | Source or reference | Identifiers | Additional information |
|---|---|---|---|---|
| Antibody | Mouse monoclonal anti GAPDH | Calbiochem | CB1001 | (1:10000) |
| Antibody | Mouse monoclonal anti PARP (cleaved) | Cell Signaling | 9541S, RRID:AB_2160592 | (1:1000) |
| Antibody | Rabbit polyclonal anti Fam20C | *Tagliabracci et al., 2013 Tagliabracci et al., 2013* | | (1:5000) |
| Chemical compound, drug | Trizol | Thermo Fisher | 15596026 | |
| Chemical compound, drug | Esmolol | abcam | ab146018 | |
| Chemical compound, drug | Dobutamine | abcam | ab120768 | |
| Chemical compound, drug | Thapsigargin | Thermo Fisher | 586005–1 MG | |
| Chemical compound, drug | $^{32}$P-ortho phosphate | PerkinElmer | NEX05 3010MC | |
| Chemical compound, drug | Gamma $^{32}$P ATP | PerkinElmer | BLU002 Z001MC | |
| Chemical compound, drug | Fura-2, AM, cell permeant | Thermo Fisher | F1221 | |
| Software, algorithm | Graph Pad Prism | | Version 7 | |
| Sequence-based reagent | RT-qPCR primers | this paper | *Table 1* | |
| Genetic reagent (*M. musculus*) | *Fam20c* Fl/+ | inGenious Targeting Laboratory | | |
| Commercial assay or kit | iScript cDNA Synthesis kit | BioRad | 1708891 | |
| Commercial assay or kit | nucleospin RNA kit | MACHEREY-NAGEL | 740955.25 | |
| Commercial assay or kit | ApopTag Peroxidase In Situ Apoptosis Detection | Millipore Sigma | S7100 | |
| Cell line (*homo sapiens*) | HEK293T | *Tagliabracci et al., 2013 Tagliabracci et al., 2013* | RRID:CVCL_6910 | |
| Cell line (*homo sapiens*) | HeLa | *Hirve et al., 2018* | RRID:CVCL_0030 | |
| Cell line (*homo sapiens*) | U2OS (WT/shFam20C) | *Tagliabracci et al., 2013 Tagliabracci et al., 2013* | RRID:CVCL_0042 | |
| Cell line (*rattus norvegicus*) | H9C2 | ATCC | CRL-1446, RRID:CVCL_0286 | |

## Mouse model

*Fam20c* $^{Fl/+}$ mice were generated by inGenious Targeting Laboratory (Ronkonkoma, NY). Briefly, exons 6–10 were flanked by two LoxP sites in the *Fam20c*-targeting vector. A neomycin-resistant gene cassette located within the two LoxP sites was flanked by two Frt sites. The targeting vector was subsequently electroporated into ES cells. Targeted iTL BA1 (C57BL/6 × 129/SvEv) hybrid embryonic stem cells were microinjected into C57BL/6 blastocysts. Resulting chimeras with a high

**Table 1.** qRT-PCR primers.

*m* corresponds to mouse, *h* corresponds to human.

| Gene | Forward | Reverse |
|------|---------|---------|
| *m*ANP | TCGTCTTGGCCTTTTGGCT | TCCAGGTGGTCTAGCAGGTTCT |
| *m*BNP | AAGTCCTAGCCAGTCTCCAGA | GAGCTGTCTCTGGGCCATTTC |
| *m*β-MCH | ATGTGCCGGACCTTGGAAG | CCTCGGGTTAGCTGAGAGATCA |
| *m*Col1a1 | GAGAGAGCATGACCGATGGATT | GCTACGCTGTTCTTGCAGTGAT |
| *m*Col 3a1 | CAGCAGTCCAACGTAGATGAATTG | CATGGTTCTGGCTTCCAGACA |
| *m*α-SMA | TCCTGACGCTGAAGTATCCG | GGCCACACGAAGCTCGTTAT |
| *m*CTGF | AATCTCCACCCGAGTTACCA | AACTTAGCCCTGTATGTCTTCAC |
| *m*Fam20C | AACATGGATCGGCATCACTAC | AGGAGCGAGAATGGAAAGC |
| *m*GAPDH | CACCATCTTCCAGGAGCGAG | CCTTCTCCATGGTGGTGAAGAC |
| *h*CHOP | GTCTAAGGCACTGAGCGTATC | CAGGTGTGGTGATGTATGAAGA |
| *h*GAPDH | ACATCGCTCAGACACCATG | TGTAGTTGAGGTCAATGAAGGG |

DOI: https://doi.org/10.7554/eLife.41378.014

percentage of agouti coat color were mated to C57BL/6 FLP mice to remove the Neo cassette. Resulting *Fam20c*$^{Fl/+}$ mice were further backcrossed onto a C57BL/6 background for greater than four generations. Tail DNA was analyzed for genotyping using the following primers: F: 5'-TTATC TGCTCATGTGCGTATGTGG-3'; R: 5'-TACAGAGCAGAACTCCAGCCACTG- 3'.

Transgenic mice expressing Cre recombinase under the control of the α-myosin heavy chain promoter (*Agah et al., 1997*) (α-*myh6*) on a C57BL/6 background were genotyped using tail DNA with the following primers: F: 5'-GCGGTCTGGCAGTAAAAACTATC-3'; R: 5'-GTGAAACAGCATTGCTG TCACTT-3'. Only male mice were used for experiments. WT and cKO mice were littermates and cage mates. Researchers performing tests and collecting data were blinded during experiments. Mice were housed in a specific pathogen-free facility with a 12 hr light and a 12 hr dark cycle, and given free access to food and water. All animal uses were approved by the Institutional Animal Care and Use Committee (IACUC) at the University of California-San Diego.

## Induced pressure overload models

For Transverse Aortic Constriction (TAC), Adult mice were anesthetized with ketamine (50 mg/kg) and xylazine (5 mg/kg) by intraperitoneal (IP) injection for initial induction and then isoflurane (0.75–1.5%) for complete induction of anesthesia. The chest cavity was entered in the second intercostal space at the left upper sternal border and the transverse aorta was isolated between the carotid arteries. Aortic constriction was performed by tying a 7–0 silk suture ligature against a 27–27.5-gauge needle (according to the body weight), and the needle was then promptly removed to yield a constriction of about 0.4 mm in diameter. Following the constriction procedure, the chest was closed with 6–0 silk sutures. Buprenorphine (0.1 mg/kg) (100 µl/mouse) was given 15–30 min prior to anticipated recovery.

For Isoproterenol infusion, mice were anesthetized with 5% isoflurane for 45 s and then maintained at 0.5% isoflurane throughout the implantation of the Alzet osmotic pumps (Durect Corp, Cupertino, CA). Mice received 30 mg/kg/day of isoproterenol for 14 days.

## Invasive hemodynamic analysis

A 1.4F micromanometer catheter (Millar Inc., Houston, TX) was inserted retrograde into the aorta via the right carotid artery and advanced into the left ventricle in anesthetized mice (100 mg/kg of ketamine and 10 mg/kg of xylazine), intraperitoneal (IP) that were intubated (100–110 strokes/minute, 0.04–0.05 ml/stroke volume). A femoral vein was cannulated with a stretched PE50 tubing for drug administration. Baseline pressure measurements were obtained. Then, graded dobutamine doses of 0.75, 2, 4, 6, and 8 µg/kg/min, or graded Esmolol doses of 10, 20, and 30 mg/kg/min, were delivered using an infusion pump (PHD 2000, Harvard Apparatus, Holliston, MA) for 3 min at each dose. Data were reported after bilateral vagotomy. LV hemodynamic parameters were recorded and

analyzed using LabChart (ADInstruments, Inc., Colorado Springs, CO), including peak LV pressure, LV end-diastolic pressure (EDP), LV dP/dt max (an index of myocardial contractility), LV dP/dt min, and Tau (time constant of LV relaxation).

## Transthoracic echocardiography

Prior to echocardiography, a depilatory cream was applied to the anterior chest wall to remove the hair. Mice were anesthetized with 5% isoflurane for 15 s and then maintained at 0.5% throughout the echocardiography examination. Small needle electrodes for simultaneous electrocardiogram were inserted into one upper and one lower limb. Transthoracic echocardiography (M-mode and 2-dimensional echocardiography) was performed using the FUJIFILM VisualSonics Inc., Vevo 2100 high-resolution ultrasound system with a linear transducer of 32-55MHz. Measurements of chamber dimensions and wall thicknesses were performed. Percentage fractional shortening (%FS) was used as an indicator of systolic cardiac function.

## Adult cardiomyocyte isolation

The hearts were removed following isoflurane anesthesia and rinsed in Krebs-Henseleit buffer B (KHB-B) (118 mM NaCl, 4.8 mM KCl, 25 mM HEPES, 1.25 mM $K_2HPO_4$,

1.25 mM $MgSO_4$, 11 mM glucose, pH 7.4). The hearts were cannulated through the aorta and perfused on a Langendorff apparatus with a KHB solution (3 ~ 5 min, 37°C), then a KHB enzyme solution (1.5 mg/ml Collagenase Type 2, 25 μM Blebbistatin) for 12 min at 37°C. After digestion, the hearts were perfused with 5 ml KHB solution to wash out collagenase. Then the hearts were minced in KHB solution, gently agitated, then filtered through a 100 μm polyethylene mesh. After settling, cells were washed once with KB solution, and stored in KB solution at room temperature before use.

## Cell shortening/re-lengthening and $Ca^{2+}$ transient studies

Adult cardiomyocytes were loaded with Fura-2-AM (1.0 μM, 20 mins), washed twice, and continuously perfused with Tyrode solution maintained at 37°C. Simultaneous measurement of intracellular $Ca^{2+}$ ($[Ca^{2+}]_i$) and cell contractility was assessed by using a video-based edge-detection system (IonOptix, Milton, MA). Briefly, cardiomyocytes were field stimulated at a frequency of 0.5 Hz using a pair of platinum wires placed on the opposite sides of the dish chamber and connected to a MyoPacer Field Stimulator (IonOptix). The cardiomyocytes being studied were displayed on the computer monitor using an IonOptix MyoCam camera. 15–25 individual myocytes were recorded and analyzed for each heart. The ratio of Fura-2 fluorescence at 340 nm and 380 nm (R) was calculated and the amplitude of intracellular $Ca^{2+}$ transient was determined by the change between the basal and peak ratio (ΔR). The amplitude of cell contraction was assessed by % change of peak shortening, and the rate of cell relaxation was assessed by the time to 63% re-lengthening (Tau).

### Cloning

Human cDNAs were purchased from either Open Biosystems or DNASU. For transient expression in mammalian cells, full-length CSQ2, Stim1, triadin, Fam20C, calumenin, calreticulin, and sarcalumenin, were cloned into pCCF with a C-terminal Flag tag. Fam20C (WT and D478A) was cloned into pCDNA3 with a C-terminal HA-tag. For recombinant expression in *E. coli*, CSQ2 (19–399) and Stim1 (22-201) were cloned into pET28 vector with a C-terminal 6X-His tag. For insect cell expression, Fam20C (93–584) was cloned into a modified pI-secSUMOstar vector (LifeSensors, Malvern, PA), in which the original SUMO tag was replaced by a MBP tag and a tobacco etch virus (TEV) protease site as previously described (*Xiao et al., 2013*). Site-directed mutagenesis was performed using QuikChange (Agilent Technologies, Santa Clara, CA). All the constructs were verified by DNA sequencing.

### Protein expression and purification

MBP-tagged Fam20C protein was expressed in Hi-5 insect cells and was purified as described previously (*Xiao et al., 2013*). The MBP tag was removed via gel filtration chromatography following TEV protease cleavage. Indicated Flag-tagged constructs were transiently expressed in HEK293 or H9C2 cells and were immunopurified from cell lysates as previously described (*Pollak et al., 2017*).

His-CASQ2 and His-Stim1 were expressed in *E. coli* BL21 (DE3) –RILP cells. Cultures were grown at 37°C to an $OD_{600}$ of 0.6–0.8 and expression was induced by addition of isopropyl-β-D-1-thiogalactopyranoside (IPTG) to 400 μM. Induced cultures were grown at 25°C overnight and cells were harvested by centrifugation at 4,680 × g for 15 min. Cell pellets were suspended in lysis buffer (50 mM Tris-HCl pH 8.0, 500 mM NaCl, 15 mM imidazole, 10% glycerol, 0.1% Triton X-100, and 10 mM β-mercaptoethanol) and lysed via sonication. The lysate was cleared by centrifugation at 11,950 × g for 45 min. The cleared lysate was incubated with Ni-NTA agarose (Invirogen) for 20 min on ice and poured through a gravity column. The beads and bound protein were washed with wash buffer (50 mM Tris-HCl pH 8.0, 500 mM NaCl, 15 mM imidazole, 10 mM β-mercaptoethanol). Protein was eluted from the column in wash buffer containing 400 mM imidazole.

## Mammalian cell culture, transfection, and $^{32}P$ orthophosphate metabolic labeling

HEK293 cells, U2OS cells, and H9C2 rat cardiomyoblast cells were cultured in Dulbecco's modified Eagle's medium (DMEM; Life Technology) supplemented with 10% FBS (Life Technology) and 100 μg/ml penicillin/streptomycin (Life Technology) at 37°C in a 5% CO2 incubator. The identity of the human cell lines were confirmed by STR profiling (Sanford Burnham Prebys Medical Discovery Institute, La Jolla) and all the cells tested mycoplasma-free via PCR. Transfection was carried out by using FuGENE-6 (Promega, Madison, WI) following the manufacturers' instructions. For metabolic radiolabeling experiments, HEK293 or H9C2 cells were seeded at $5 \times 10^5$ cells per well in 6-well plate; 20 hr later, cells were transfected with the indicated plasmids. 1 day after transfection, metabolic labeling was started by replacing the medium with phosphate-free DMEM containing 10% dialyzed FBS and 1 mCi/ml $^{32}P$ orthophosphate (PerkinElmer, Waltham, MA). After labeling for 8 hr, the cell lysate was collected and the cell debris was removed by centrifugation. Flag-tagged proteins were immunoprecipitated from the supernatant and analyzed for protein and $^{32}P$ incorporation by immunoblotting and autoradiography.

## In vitro phosphorylation assays

For the in vitro kinase assays substrates were purified from bacteria and Fam20C was purified from insect cells, and the proteins were incubated in 50 mM Tris pH 7.0, 10 mM $MnCl_2$, 1 mM [γ-$^{32}P$] ATP (specific activity = 100–500 cpm/pmol), 0.25 mg/mL substrate, and 10 μg/mL Fam20C at 30° C as previously described (*Tagliabracci et al., 2012*). Reactions were terminated at the indicated time points by SDS loading buffer, 15 mM EDTA, and boiling. Reaction products were separated by SDS-PAGE and incorporated radioactivity was visualized via autoradiography and immunoblotting.

## Histology

Hearts were isolated from mice, washed with PBS and fixed for 24 hr in a 10% Formalin. The hearts were then placed in 70% ethanol and were submitted to the University of California, San Diego histology core for paraffin embedding and sectioning for H and E and Masson's Trichrome staining. The Tunel assay was performed using the ApopTag Peroxidase In Situ Apoptosis Detection Kit per the manufacturer's instructions. Slides were scanned via the AT2 Aperio Scan Scope. Masson's Trichrome and Tunel stained slides were analyzed in five high powered fields (HPF) per sample by a histologist blinded to genotype.

## Turbidity assay

For the turbidity assay, His-CASQ2 was phosphorylated by direct addition of ATP, $MnCl_2$, and FAM20C to the eluted sample (above). The final reaction conditions contained 50 mM Tris-HCl pH 8.0, 500 mM NaCl, 400 mM imidazole, 10 mM β-mercaptoethanol, 1 mM ATP, 1 mM $MnCl_2$, and ~1 μg/mL FAM20C. Saturated phosphorylation (>95%) was demonstrated by removing a small portion of the reaction, adding radioactive ATP [γ-$^{32}P$], and observing a plateau of time-dependent incorporation of radioactive phosphate via SDS-PAGE and autoradiography (not shown). The unphosphorylated protein was treated in tandem under identical conditions but excluding Fam20C. Reactions were quenched after 4 hr by the addition of EDTA to 10 mM.

The phosphorylated and control protein was dialyzed against TG buffer (10 mM Tris-HCl pH 7.0%–10% glycerol). Samples were concentrated to 1 mL and purified further with size exclusion

chromatography. The protein was loaded onto a HiLoad 16/600 Superdex 200 pg column (GE Healthcare) and eluted in TG buffer. Elution was monitored by absorbance at 280 nm and peak fractions were analyzed with SDS-PAGE followed by Coomassie blue staining. Fractions containing homogenous CASQ2 were pooled and used for further analysis.

Turbidity assays were performed by combining either phosphorylated or unphosphorylated His-CASQ2 protein with $CaCl_2$ in TG buffer at a 1:1 ratio to yield final reaction mixtures containing 4.5 µM CASQ2, 10 mM Tris-HCl pH 7.0, 10% glycerol, and $CaCl_2$ at the concentrations indicated. The mixtures were incubated at room temperature for 30 min with gentle shaking and 200 µL of the individual reactions were transferred to a transparent Nuclon 96-well plate (Thermo). The turbidities of the samples were determined by measuring the absorbance at 350 nM in Infinite M200 Pro plate reader (Tecan).

## Single-cell Ca$^{2+}$ influx assay

Single-cell Ca2 +imaging was performed using HeLa cells that had been transfected with either Fam20C WT or Fam20C KI plasmid, and non-transfected HeLa cells. The cells were loaded with 5 µM Fura-2-acetoxymethyl ester for 45–60 min at 37°C in DMEM containing 0.02% Pluronic F-127 and 10 mM HEPES pH 7.4, washed twice with fresh media, and analysed immediately. Modified Ringer's solution used in this assay consisted of 125 mM NaCl, 5 mM KCl, 1.5 mM MgCl2, 10 mM D-glucose, and 20 mM HEPES (pH 7.4 with NaOH), with the addition of 2 mM $CaCl_2$, or 1 µM thapsigargin (TG), where indicated. Coverslips were assembled into a chamber on the stage of an Olympus IX 71 microscope equipped with an Olympus UPLSAPO 20×, NA 0.75, objective. Cells were alternately illuminated at 340 nm and 380 nm with the Polychrome V monochromator (TILL Photonics) using ET - Fura2 filter (Chroma Technology Corp., catalog number 79001). The fluorescence emission at $\lambda > 400$ nm (LP 400 nm, Emitter 510/80 nm) were captured with a CCD camera (SensiCam, TILL Imago). Ratio images were recorded at intervals of 4 s. $Ca^{2+}$ concentration was estimated from the relation (*Grynkiewicz et al., 1985*): $[Ca^{2+}]_i = K_d ((R-R_{min})/(R_{max}-R)) (S_{f2}/S_{b2})$, where $K_d = 220$ nM, and the values of $R_{min}$, $R_{max}$, and $(S_{f2}/S_{b2})$ were determined from an in situ calibration of Fura-2 in HeLa cells. Data were analyzed using TILL Vision (TILL Photonics).

## Stim1 Ca$^{2+}$ concentration-dependent crosslinking assay

HeLa cells were co-transfected with Stim1(A230C) and Fam20C WT, or with Stim1(A230C) and Fam20C KI plasmids. The crosslinking was performed as described previously (*Hirve et al., 2018*). Briefly, the cells ($\sim 12 \times 10^6$) were scraped from the substrate and resuspended in Chelex-treated dilution buffer with no added EGTA or $Ca^{2+}$. Cellular membranes were prepared and apportioned to twelve wells containing resuspension buffer supplemented to give either a final EGTA concentration of 0.5 mM or final $Ca^{2+}$ concentrations ranging from 0.3 µM to 2 mM. Iodine oxidation, nonreducing SDS-PAGE analysis, and immunoblotting using anti-Stim1 antibody were performed as described previously (*Hirve et al., 2018*).

## ER Ca$^{2+}$ handling

ER $Ca^{2+}$ release was quantified following treatment with 2 µM Thapsigargin induced ER Ca release using the dual wavelength ratiometric cytosolic Ca dye, Fura2-AM (Invitrogen), on the FLEX-Station (Molecular Devices) (*Marshall et al., 2006*) in a 96-well plate format with 3–6 replicates per sample, as previously described (*Wiley et al., 2013*). Minimum and maximum signal intensities were determined for each run. Ratiometric measurements were converted to intracellular $[Ca^{2+}]$ using the following equation: $[Ca^{2+}] = K_d[(R-R_{min})/(R_{max}-R)](F_{380max}/F_{380min})$.

## Cardiac fractionation

For generating the whole tissue lysate, freshly isolated hearts were homogenized in a buffer containing 50 mM Tris–HCl (pH 8.0), 150 mM NaCl, 0.5% NP-40, 10% glycerol, 0.5 mM EDTA, and a protease inhibitors cocktail. To enrich the ER/Golgi, hearts were homogenized in HME buffer containing 10 mM Tris–HC (pH 7.4), 250 mM sucrose, 1 mM EDTA, and a protease inhibitors cocktail. The homogenate was centrifuged at 1000xg to remove nuclei and unbroken cells. The supernatant was centrifuged at 3000xg to pellet heavy mitochondria. The 3000xg supernatant was then centrifuged

at 17,000xg to pellet the ER, Golgi and light mitochondria. This pellet was resuspended in HME buffer as the ER/Golgi-enriched fraction.

## Antibodies

The following antibodies were used in for protein immunoprecipitation or immunoblotting: mouse anti-Flag M2 (Sigma), rabbit anti-Flag (Sigma), anti-GAPDH (Calbiochem), rabbit anti-Stim1 (Cell Signaling), anti-Bip/GRP78 (Abcam), anti-PARP (cleaved) (Cell Signaling), anti-Calsequestrin 2 (Thermo), anti-Fam20C (*Tagliabracci et al., 2014*), and anti-His (Invitrogen).

## Statistics

All data are shown as mean ±SEM. Replicates are indicated in figure legends, with n representing the number of experimental replicates. When comparing two groups, two tailed Student's *t* test was performed. Statistical analysis was performed using GraphPad Prism 6.0 (GraphPad Software). $p < 0.05$ was considered statistically significant.

## qRT-PCR

For qRT-PCR analysis, total RNA from cardiac tissue was isolated from using Trizol (Invitrogen) per the manufacturer's instructions. For Chop expression analysis, total RNA was isolated cells using the NucleoSpin RNA kit (MACHEREY-NAGEL, Bethlehem, PA). cDNA was synthesized using the iScript kit (Bio-Rad). qRT-PCR analysis was performed using the Power SYBR Green PCR Master Mix (Applied Biosystems, Waltham, MA) on Applied Biosystems 7500 Real-Time PCR System. Data were normalized to corresponding GAPDH levels. The indicated primers were used (*Table 1*).

## Acknowledgements

We thank the UCSD Histology Core for help with histology. We thank Melissa Barlow for assistance with animal breeding. We thank Carolyn Worby and Sourav Banerjee for valuable input.

## Additional information

### Funding

| Funder | Grant reference number | Author |
|---|---|---|
| National Institutes of Health | F32HL136122 | Adam J Pollak |
| National Institutes of Health | 3T32HL007444-34S1 | Adam J Pollak |
| National Institutes of Health | 5T32CA009523-32 | Joshua E Mayfield |
| National Institutes of Health | AI109842 | Patrick G Hogan |
| National Institutes of Health | AI040127 | Patrick G Hogan |
| National Institutes of Health | DK018849-41 | Jack E Dixon |
| National Institutes of Health | DK018024-43 | Jack E Dixon |
| National Institutes of Health | R37HL028143 | Joan Heller Brown |

The funders had no role in study design, data collection and interpretation, or the decision to submit the work for publication.

### Author contributions

Adam J Pollak, Conceptualization, Data curation, Funding acquisition, Methodology, Writing—original draft, Writing—review and editing; Canzhao Liu, Patrick G Hogan, Conceptualization, Data curation; Aparna Gudlur, Yusu Gu, Conceptualization, Data curation, Investigation; Joshua E Mayfield, Conceptualization and Data curation; Nancy D Dalton, Ju Chen, Conceptualization, Data curation, Methodology; Joan Heller Brown, Conceptualization, Resources, Writing—review and editing; Sandra E Wiley, Conceptualization, Data curation, Writing—original draft, Writing—review and editing;

Kirk L Peterson, Conceptualization, Writing—review and editing; Jack E Dixon, Conceptualization, Data curation, Funding acquisition, Writing—original draft, Writing—review and editing

## Author ORCIDs
Adam J Pollak (iD) http://orcid.org/0000-0001-5266-4691
Jack E Dixon (iD) http://orcid.org/0000-0002-8266-5449

## Ethics

Animal experimentation: This study was performed in strict accordance with the recommendations in the Guide for the Care and Use of Laboratory Animals of the National Institutes of Health. All of the animals were handled according to approved institutional animal care and use committee (IACUC) protocols of the University of California at San Diego. The protocol was approved by the Committee on the Ethics of Animal Experiments of the University of California at San Diego (Protocol Number: S03039). All surgery was performed under ketamine and xylazine anesthesia, and every effort was made to minimize suffering.

## Decision letter and Author response

Decision letter https://doi.org/10.7554/eLife.41378.017
Author response https://doi.org/10.7554/eLife.41378.018

# Additional files

## Supplementary files
• Transparent reporting form
DOI: https://doi.org/10.7554/eLife.41378.015

## Data availability
All data generated or analysed during this study are included in the manuscript and supporting files.

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
