## [Decision Letter]

Thank you for submitting your article "A Secretory Pathway Kinase Regulates Sarcoplasmic Reticulum Ca^2+^ Homeostasis and Protects Against Heart Failure" for consideration by *eLife*. Your article has been favorably reviewed by three peer reviewers, and the evaluation has been overseen by John Kuriyan as the Reviewing Editor and Senior Editor. The following individual involved in review of your submission has agreed to reveal their identity: John D Scott (Reviewer #2).

The reviewers have discussed the reviews with one another and the Reviewing Editor has drafted this decision to help you prepare a revised submission.

This manuscript reports comprehensive analyses of the role of Fam20C, a relatively novel ER/SR lumen-localized protein kinase, in cardiac function. The authors show that FAM20C can phosphorylate calreticulin, calumenin, sarcalumenin and calsequestrin. A dead FAM20C, made by mutating aspartate 478 to an alanine, can no longer phosphorylate calsequestrin. Triadin phosphorylation, in contrast, is not abolished. The phosphorylation site on calsequestrin is S385, which is conserved in mammals. Phosphorylated CSQ2 aggregates more easily. Similarly, STIM1 is phosphorylated on S88 by FAM20C both in vitro and in vivo. FAM20C KO mice experience heart failure upon exposure to chemical stressors as well as aging. Their hearts show hypertrophy and fibrosis. The authors show that FAM20C activity combats cardiac stress in mice, their proposed mechanism involves changes to cardiac contractility and calcium transients and differential phosphorylation or ER proteins.

In general, the findings are interesting, supporting an important and novel pathophysiological functions for protein phosphorylation within the SR in regulating cardiac calcium handling and contractility. The manuscript is well suited for publication in *eLife*.

The reviewers raise some concerns that are noted below. While additional data that address these issues would be useful, the authors can also choose to revise the manuscript in a way that makes clear the limitations in interpretation, without providing new data. In addition, the reviewers have provided a number of suggestions that could improve the manuscript. Please consider all of these in revising the manuscript.

Important points to address:

1) While the authors show that Fam20C can phosphorylate calsequestrin-2 (cardiac-specific isoform), STIM1, and other ER/SR proteins, the current data provide only a limited biochemical hint as to how this may impact calcium handling by the SR/ER. The loss of Fam20C appears to inhibit calsequestrin-2 oligomerization, which is predicted to decrease SR calcium storage. While data from HELA cells in Figure 8 appears consistent with decreased ER calcium storage, these cells likely express calsequestrin-1, not calsequestrin-2. Does Fam20C also regulate calsequestrin-1? Can the authors obtain evidence for decreased calcium storage in cardiomyocytes? Similarly, biochemical data indicate that Fam20C drives STIM1 dimerization (activation) at high calcium concentrations, which is predicted to result in store-operated calcium entry via Orai1 even at high calcium. Does the loss of Fam20C affect SOCE in cardiomyocytes or other cells? Without these functional data, there is some concern that the authors are over-interpreting the very indirect biochemical changes as important regulatory effects of Fam20C. Does higher turbidity necessarily correlate with formation of higher order oligomers and increased calcium binding to calsequestrin? Similarly, does the dimerization of STIM1 necessarily correlate with activation? Without such functional data, the mechanistic link between the loss of Fam20C and the cardiac phenotypes is rather unclear.

2) It appears from the manuscript that the data reporting and statistical analyses do not meet current expectations for transparency and rigor in several ways. Notably, the authors do not provide any explanation of their approach to data analysis using *eLife*'s supplementary transparent reporting form. First, there is no indication that several qualitative data sets (e.g., gels and blots in Figures 1A, C, D; Figures 2B-E) have been replicated. This is particularly problematic for the analysis of the STIM1-S88G mutant in Figure 2C, D, where the qualitative differences are not very striking/convincing, especially if the phosphorylation is "eyeball" normalized to levels of total protein on the gels. These studies need to be repeated using more quantitative approaches to show that there is in fact a significant difference. Second, the authors report mean ± SEM data using bar graphs in several panels (e.g., Figure 3C, D), which can be very misleading. They should superimpose the individual data points on these bar graphs to provide a transparent indication of data variability (as they have done in several other panels), or they should use box/whisker graphs to depict data variability in cases where there are larger numbers of data points. Third, the statistical analyses are not adequate, and as such do not convincingly support the authors interpretations. It appears that the authors exclusively use t-tests to compare selected pairs of data sets, whereas the experiments are comparing data across two variable (genotype and age), such that a 2-way ANOVA is needed. We do not think this will have a big impact on interpretation of many data sets, but in some cases it might. For example, multiple panels of Figure 5 are not very convincing because it appears that a single "outlier" point may be driving a difference. Was an outlier test performed? Were data tested for a normal distribution? Similarly for Figure 4A at 9 months of age. Fourth, how were sample sizes determined for the in vivo studies? Fifth, all the gels and blots need to be labeled to indicate the positions of molecular weight markers.

Other points to consider:

1) The designation FAM20C DA for the kinase dead form of the enzyme is jargon. We suggest using a more clearly understood abbreviation.

2) How many times was the turbidity assay in Figure 1E performed. Since the effect of phosphorylation on calsequestrin 2 is modest this information should be included in the body of Figure 1.

3) The authors state in the legend for Figure 2E that "Crosslinked, dimerized (active) Stim 1 is indicated by the upper arrow, and monomeric (inactive) Stim1 is indicated by the lower arrow". The reviewers feel that it would be more accurate to say that crosslinked Stim1 is multimerized. There is no real biochemical proof that the complex is only a dimer.

4) The mouse studies in Figure 4 are nice but the authors need to clearly indicate that red traces represent Fam20 -/- animals and that black traces represent floxed controls. This is an important point that complicates interpretation of the data. (please note that Figures 5 and 6 provide this information).

5) The in vivo data presented in this article are impressive and strongly support a role for FAM20C in cardiovascular disorders. Many of the effects that are shown in the physiological analysis are reminiscent of changes in the β adrenergic and other second messenger signaling pathways that cause heart disease. While these effects are clearly distinct from the findings of this article there may be some merit in adding something to the Discussion that puts the new findings of this work in context with this well-established literature.

6) For the experiments in Figure 1A, please clarify whether all the over-expressed proteins are exclusively localized with the ER, or does the over-expression result in some leak into the cytoplasm (which might impact the results).

7) In Figure 1B, the authors should indicate whether the highlighted S-X-E sequence motif is the only such conserved motif in calsequestrin-2.

8) In Figure 1C, it seems odd that the S385A mutant protein migrates more slowly than the WT because phosphorylation more typically reduces mobility. Is the gel correctly oriented?

9) The Stim1(A230C) dimerization assay in Figure 2E detecting supposedly activated dimeric confirmations of Stim1 could be explained in more detail for a diverse readership.

10) The title for Figure 4—figure supplement 1 is not phrased very well: it may be better written as: "Role of Fam20C in the acute hemodynamic response to β-adrenergic receptor activation."

11) In Figure 3, the meaning of the red/black traces/bars does not appear to be explained.

12) Figure 1E. Turbidity is a poor proxy for oligomerization but a good proxy for precipitation or polymerization. Maybe you want to call it "polymerization" or "aggregation" according to Wang 1998 Nat. Struct. Biol. 5, 476-483. Shin, Ma and Kim, 2000, shows interaction of CSQ with triadin but not CSQ oligomerization.

13) Figure 2E. WB image is either overexposed or at high contrast. Increase in Ca^2+^ and phosphorylation encourages STIM1 dimerization.

14) Figure 3. Please state red (FAM20C KO) and black (WT) in figure key or legend.

---

## [Author Response]

Important points to address:1) While the authors show that Fam20C can phosphorylate calsequestrin-2 (cardiac-specific isoform), STIM1, and other ER/SR proteins, the current data provide only a limited biochemical hint as to how this may impact calcium handling by the SR/ER. The loss of Fam20C appears to inhibit calsequestrin-2 oligomerization, which is predicted to decrease SR calcium storage. While data from HELA cells in Figure 8 appears consistent with decreased ER calcium storage, these cells likely express calsequestrin-1, not calsequestrin-2. Does Fam20C also regulate calsequestrin-1? Can the authors obtain evidence for decreased calcium storage in cardiomyocytes? Similarly, biochemical data indicate that Fam20C drives STIM1 dimerization (activation) at high calcium concentrations, which is predicted to result in store-operated calcium entry via Orai1 even at high calcium. Does the loss of Fam20C affect SOCE in cardiomyocytes or other cells? Without these functional data, there is some concern that the authors are over-interpreting the very indirect biochemical changes as important regulatory effects of Fam20C. Does higher turbidity necessarily correlate with formation of higher order oligomers and increased calcium binding to calsequestrin? Similarly, does the dimerization of STIM1 necessarily correlate with activation? Without such functional data, the mechanistic link between the loss of Fam20C and the cardiac phenotypes is rather unclear.

Calsequestrin-1 lacks the C-terminal tail that is necessary for proper calsequestrin-2 oligomerization. Notably, this tail contains the only Fam20C S-x-E phosphorylation site on calsequestrin-2 (as noted in the third paragraph of the subsection “Fam20C phosphorylates multiple SR regulatory proteins important for Ca^2+^ homeostasis”). Interestingly, calsequestrin-1 has a Fam20C site in an alternative location on the protein. While future work is necessary to determine if and how Fam20C regulates calsequestrin-1, it is overwhelmingly likely that the lack of the phosphorylation site on the C-terminal tail of calsequestrin-1 will make its biochemical response to Fam20C phosphorylation different than calsequestrin-2’s biochemical response. Importantly, it is likely that Fam20C mediated phosphorylation of Stim-1 is the cause of the effects demonstrated in the Hela cells (Figure 8A).

We chose to employ the turbidity assay to study calsequestrin-2, as several studies have used this as a basis for investigations of calsequestrin-2 polymerization (Sanchez et al., 2011). As noted below, and now in the manuscript, we state that turbidity is used to study polymerization, not oligomerization (subsection “Fam20C phosphorylates multiple SR regulatory proteins important for Ca^2+^ homeostasis”, fourth paragraph). Future work involving additional experimental approaches is necessary to more completely determine Fam20C’s role in calsequestrin-2 oligomerization and calcium binding properties.

While we think that a thorough investigation of Fam20C mediated regulation of calcium storage and SOCE in cardiomyocytes will be very interesting, we feel that it is beyond the scope of our current study. We now indicate these limitations of our current study in the sixth paragraph of the Discussion. However, we’d like to note here that Hela cells have been a well-established model to study SOCE.

Our current study employs a recent but thoroughly developed method that relates Stim-1 mediated “dimerization” to Stim-1 activation (Hirve et al., 2018). In particular, we note that Fam20C mediated activation of SOCE is similar to the activation of SOCE demonstrated via Stim-1 activating mutants (Hirve et al., 2018), as described in the last paragraph of the subsection “Fam20C regulates ER Ca^2+^ homeostasis”. This is suggestive yet not definitive evidence for a broader role for Fam20C regulation of Stim1 in other cell types. This is noted in the sixth paragraph of the Discussion.

Triadin has known cytosolic phosphorylations, which like Stim-1 causes a background of phosphorylation in the KI lane in Figure 1A, as noted in the paper (subsection “Fam20C phosphorylates multiple SR regulatory proteins important for Ca^2+^ homeostasis”, second paragraph).

2) It appears from the manuscript that the data reporting and statistical analyses do not meet current expectations for transparency and rigor in several ways. Notably, the authors do not provide any explanation of their approach to data analysis using eLife's supplementary transparent reporting form. First, there is no indication that several qualitative data sets (e.g., gels and blots in Figures 1A, C, D; Figures 2B-E) have been replicated. This is particularly problematic for the analysis of the STIM1-S88G mutant in Figure 2C, D, where the qualitative differences are not very striking/convincing, especially if the phosphorylation is "eyeball" normalized to levels of total protein on the gels. These studies need to be repeated using more quantitative approaches to show that there is in fact a significant difference. Second, the authors report mean ± SEM data using bar graphs in several panels (e.g., Figure 3C, D), which can be very misleading. They should superimpose the individual data points on these bar graphs to provide a transparent indication of data variability (as they have done in several other panels), or they should use box/whisker graphs to depict data variability in cases where there are larger numbers of data points. Third, the statistical analyses are not adequate, and as such do not convincingly support the authors interpretations. It appears that the authors exclusively use t-tests to compare selected pairs of data sets, whereas the experiments are comparing data across two variable (genotype and age), such that a 2-way ANOVA is needed. We do not think this will have a big impact on interpretation of many data sets, but in some cases it might. For example, multiple panels of Figure 5 are not very convincing because it appears that a single "outlier" point may be driving a difference. Was an outlier test performed? Were data tested for a normal distribution? Similarly for Figure 4A at 9 months of age. Fourth, how were sample sizes determined for the in vivo studies? Fifth, all the gels and blots need to be labeled to indicate the positions of molecular weight markers.

We now indicate that the blots shown in Figure 1 and 2 are representative of 3 replicates in the figure legends.

The experiment done in Figure 2D was repeated with an appropriately exposed western blot to more convincingly show Fam20C mediated phosphorylation of Stim1-Ser88 in cells. In addition, we employed densitometry to more accurately quantify the phosphorylation shown in both Figure 2C and D, which is now shown as newly generated Figure 2E.

The individual data points are now shown in Figure 3C and D. 2-way ANOVA tests were performed for the relevant datasets and are now reported within the figures.

We performed outlier and normalized distribution tests for all data sets from Figures 4 and 5. All data were normally distributed via the Shapiro-Wilk test, except for Figure 4A (max contractility at 9 months, cKO). We performed the Grubbs test for outliers and all datasets contained no outliers, except for Figure 4A (max contractility at 9 months, cKO) and Figure 5A (LVIDd at 9 months, cKO). However, it has been a long-standing practice of Dr. Kirk Peterson’s laboratory to report and include all experimental data collected in a given experiment, unless a clear error was identified. We strongly feel that our physiological datasets (Figure 4 and 5 in particular) do not contain outliers, and are instead genuine representations of the variation induced as a physiological consequence of the removal of Fam20C from cardiomyocytes. Therefore, we chose to include the data as is. Regardless, removal of these outliers would not change the fundamental conclusions that Fam20C cKO mice develop heart failure and diastolic dysfunction following aging in comparison to controls.

The sample sizes were determined by following the conventions of established literature, where 8-15 mice per group were used.

The gel markers are now included.

Other points to consider:1) The designation FAM20C DA for the kinase dead form of the enzyme is jargon. We suggest using a more clearly understood abbreviation.

The label is changed to kinase-inactive (KI), which is a more established form of nomenclature to describe an inactive kinase.

2) How many times was the turbidity assay in Figure 1E performed. Since the effect of phosphorylation on calsequestrin 2 is modest this information should be included in the body of Figure 1.

The turbidity assay was repeated 3 times, as indicated in the figure legend.

3) The authors state in the legend for Figure 2E that "Crosslinked, dimerized (active) Stim 1 is indicated by the upper arrow, and monomeric (inactive) Stim1 is indicated by the lower arrow". The reviewers feel that it would be more accurate to say that crosslinked Stim1 is multimerized. There is no real biochemical proof that the complex is only a dimer.

The designation of “dimer” is now replaced with “multimer” in Figure 2.

4) The mouse studies in Figure 4 are nice but the authors need to clearly indicate that red traces represent Fam20 -/- animals and that black traces represent floxed controls. This is an important point that complicates interpretation of the data. (please note that Figures 5 and 6 provide this information).

Legends to indicate genotype are now indicated in all figures.

5) The in vivo data presented in this article are impressive and strongly support a role for FAM20C in cardiovascular disorders. Many of the effects that are shown in the physiological analysis are reminiscent of changes in the β adrenergic and other second messenger signaling pathways that cause heart disease. While these effects are clearly distinct from the findings of this article there may be some merit in adding something to the Discussion that puts the new findings of this work in context with this well-established literature.

We added a section (line 345), including new references (Ling et al., 2009 and Braz et al., 2004), to contextualize our results.

6) For the experiments in Figure 1A, please clarify whether all the over-expressed proteins are exclusively localized with the ER, or does the over-expression result in some leak into the cytoplasm (which might impact the results).

Our lab has previously over-expressed several secretory pathway proteins and monitored their locations via immunofluorescence (Tagliabracci et al., 2012; 2016). In every case, the proteins appear to be expressed only within the secretory pathway. Therefore, over-expression of proteins does not cause any leakage into the cytoplasm. Importantly, both triadin and Stim-1 are transmembrane proteins that contain cytosolic portions that have been documented as phosphorylated. This is now mentioned in the paper (subsection “Fam20C phosphorylates multiple SR regulatory proteins important for Ca^2+^ homeostasis”, second paragraph), which fully accounts for the results.

7) In Figure 1B, the authors should indicate whether the highlighted S-X-E sequence motif is the only such conserved motif in calsequestrin-2.

It is now indicated that calsequestrin-2 has a single S-x-E sequence in the protein in both the text (subsection “Fam20C phosphorylates multiple SR regulatory proteins important for Ca^2+^ homeostasis”, third paragraph) and the figure legend.

8) In Figure 1C, it seems odd that the S385A mutant protein migrates more slowly than the WT because phosphorylation more typically reduces mobility. Is the gel correctly oriented?

Yes, the gel is correctly oriented. Multiple possible explanations exist for why the protein runs slower, such as changes to the protein’s glycosylation pattern. A detailed understanding of this is beyond the scope of the current paper.

9) The Stim1(A230C) dimerization assay in Figure 2E detecting supposedly activated dimeric confirmations of Stim1 could be explained in more detail for a diverse readership.

We have added a more general description of this assay (subsection “Fam20C phosphorylates multiple SR regulatory proteins important for Ca^2+^ homeostasis”, last paragraph) and included a better description in the Materials and methods section.

10) The title for Figure 4—figure supplement 1 is not phrased very well: it may be better written as: "Role of Fam20C in the acute hemodynamic response to β-adrenergic receptor activation."

We changed the title to the one suggested here.

11) In Figure 3, the meaning of the red/black traces/bars does not appear to be explained.

It is now explained in the figure legend.

12) Figure 1E. Turbidity is a poor proxy for oligomerization but a good proxy for precipitation or polymerization. Maybe you want to call it "polymerization" or "aggregation" according to Wang 1998 Nat. Struct. Biol. 5, 476-483. Shin, Ma and Kim, 2000, shows interaction of CSQ with triadin but not CSQ oligomerization.

The nomenclature was changed from oligomerization to polymerization. A new reference (Park et al., 2003) was added to describe calsequestrin-2 oligomerization.

13) Figure 2E. WB image is either overexposed or at high contrast. Increase in Ca^2+^ and phosphorylation encourages STIM1 dimerization.

We feel that the western blot shows an appropriate range of contrast to support the conclusion that Stim1 can be activated by Fam20C under high calcium conditions.

14) Figure 3. Please state red (FAM20C KO) and black (WT) in figure key or legend.

This has been done.

As a final note, the wording of a few sections of the manuscript were slightly altered for added clarity. We also decided to slightly alter the presentation of the data in Figure 8B.